# Training the Untrainable: Introducing Inductive Bias via Representational Alignment

**Vighnesh Subramaniam[1]\*, David Mayo[1], Colin Conwell[2],**
**Tomaso Poggio[1], Boris Katz[1], Brian Cheung[1†], Andrei Barbu[1†]**
[1]MIT CSAIL, CBMM [2]Department of Cognitive Science, Johns Hopkins University
[1]{vsub851,dmayo2,tp,boris,cheungb,abarbu}@mit.edu
[2]cconwel2@jhu.edu

## Abstract

We demonstrate that architectures which traditionally are considered to be ill-suited for a task can be trained using inductive biases from another architecture. We call a network untrainable when it overfits, underfits, or converges to poor results even when tuning their hyperparameters. For example, fully connected networks overfit on object recognition while deep convolutional networks without residual connections underfit. The traditional answer is to change the architecture to impose some inductive bias, although the nature of that bias is unknown. We introduce guidance, where a guide network steers a target network using a neural distance function. The target minimizes its task loss plus a layerwise representational similarity against the frozen guide. If the guide is trained, this transfers over the architectural prior and knowledge of the guide to the target. If the guide is untrained, this transfers over only part of the architectural prior of the guide. We show that guidance prevents FCN overfitting on ImageNet, narrows the vanilla RNN–Transformer gap, boosts plain CNNs toward ResNet accuracy, and aids Transformers on RNN-favored tasks. We further identify that guidance-driven initialization alone can mitigate FCN overfitting. Our method provides a mathematical tool to investigate priors and architectures, and in the long term, could automate architecture design.
Project website at `https://untrainable-networks.github.io`

## 1 Introduction

When creating neural networks, as a community, we follow recipes that select among a few architectures that are known to work for particular tasks [63, 12, 25]. Architecture is critical, encoding essential inductive biases i.e. priors that profoundly impact their learning capabilities and performance across various tasks. Convolutional nets revolutionized vision [47, 31], and Transformers reshaped language [74, 20, 1]. Despite this, architecture design is a dark art because the precise relationship between architectures and the priors they impose is poorly understood. For example, there is discussion about exactly what the role of residual connections is [39]. This reflects a broader challenge: we rarely understand exactly what inductive biases our architectures encode. Our lack of understanding makes architecture design challenging. Given new application spaces for neural networks with rising compute costs like inference-time scaling [56], this challenge has become even more relevant.

Recent theorems [59] state that for each function which is efficiently Turing computable, there exists a deep network that can approximate it well. Furthermore, a graph representing such a function is

---

\*Corresponding author.
†Equal senior contribution

39th Conference on Neural Information Processing Systems (NeurIPS 2025).

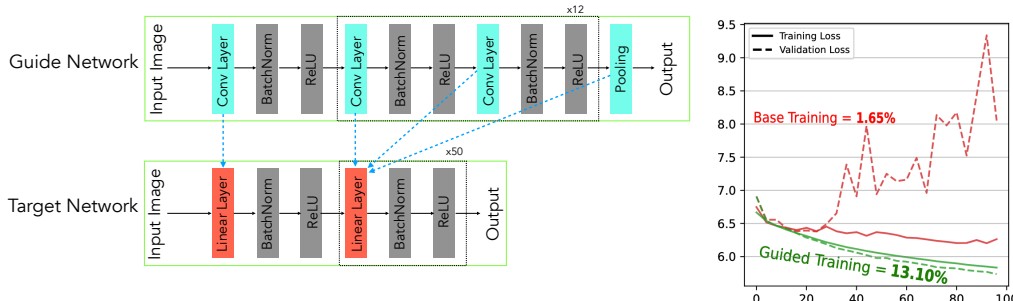

Figure 1: **Guidance makes untrainable networks trainable via representational similarity**. Given a target which cannot be trained effectively on a task, we train this target with a layerwise representational-alignment term against a fixed guide—trained or random—which remains unchanged during task training. This transfers only the guide's architectural bias, turning a network that would otherwise overfit or underfit into one that learns effectively (e.g., a deep FCN guided by a random ResNet for image classification).

compositionally sparse, that is the nodes of the associated Directed Acyclic Graph (DAG) represent constituent functions with a small effective dimensionality. A reasonable conjecture is that neural networks with an architecture which is similar to the DAG of the unknown target function are especially successful in learning it, as it is the case for convolutional networks for image recognition and similar tasks. However, empirically testing or transferring those structural priors remains an open challenge. Because we do not understand the relationship between the kinds of priors on the target functions that different architectures impose, even simple questions have no known answer. For example, can an FCN's initialization be tailored to mimic a CNN's inductive bias, despite their distinct graphs?

To bridge this gap, we introduce a novel empirical tool, *guidance*. Given a *target network*, we guide it with a *guide network*. In addition to the target's original loss, the target attempts to match the representation of its intermediate layers to those of the guide. We use a measure of representational similarity [45, 17, 16], also termed a neural distance function, to compute the distance between representations of two arbitrary layers. Neural distance functions are often used in neuroscience to compare activity in networks and brains [68, 14, 71]. In light of recent work that shows that networks of very different architectures have internal activity that is extremely similar to one another [30, 14, 15], we repurpose this distance function as a means to transfer priors between networks layer by layer. Surprisingly, even a randomly initialized guide—incapable of solving the task—yields large performance gains, proving architectures alone encode powerful priors. This surprising finding demonstrates that neural architectures alone, independent of parameter training, impose meaningful inductive biases that are useful for downstream tasks.

We make the following contributions:

1. We develop guidance to transfer priors between networks using representational alignment and investigate one representational alignment method, centered kernel alignment, CKA [45].
2. We empirically differentiate between architectural and trained inductive biases, showing that architectural priors alone can significantly improve network performance. This underscores the intrinsic structural power of architectures independent of learned parameters.
3. We show that RNNs significantly improve their copy-and-paste accuracy when guided by a Transformer. Transformers increase their parity accuracy when guided by an RNN. RNNs close much of the gap to a Transformer on language modeling when guided by one.
4. We show that deep or wide fully connected networks stop overfitting when guided by a ResNet. No-skip CNNs close much of the gap with ResNets when guided by a ResNet. Fully-connected networks stop overfitting with guidance-only initialization schemes.

Our method provides a powerful empirical tool to further theoretical insights in architectural design. Guidance enables systematic investigation of the structural foundations of successful architectures and clarifies the distinction between architectural and trained inductive biases. Minimizing CKA lets

you clamp any subset of a target network's activations onto those of a frozen guide network, sweep that clamp across architectures and priors, and observe what fails or succeeds. Guidance is a knob you can turn to inject architectural priors at will, something one cannot do with cross-entropy or weight-decay alone.

Our work has a number of limitations. We aimed for coverage of many tasks instead of maximal performance on any one task. This would have required us to carefully tune the hyperparameters involved. We preferred to show how guidance works in general rather than in cherry picked and carefully tuned settings. To that end, we also did not optimize networks to convergence, nor did we attempt to experiment with other optimizers. Once we reproduced a well-known problematic training phenomenon, we showed that it could be overcome. We consider a network trainable and a problem to be overcome when the original problem disappears. For example, successfully training fully connected networks for object recognition was hopeless because they immediately overfit; using our guidance method they no longer do so. This does not mean that they are necessarily useful as object recognizers at present. In the case of fully connected networks, their present performance with guidance training is too low, but with additional work we believe their performance could be substantially increased now that their train and test loss are moving in the right direction. In some cases, by applying guidance, we do see large useful improvements, such as with RNNs and Transformers, as well as deep CNNs, although much more remains to be exploited there too.

## 2   Related Work

**Representational Distance**: Our method builds on several metrics that measure distance between high-dimensional activations extracted from neural networks or activity in the brain [44]. Some of these distance metrics make comparisons based on kernel matrices [45, 17, 16] or relative distances [46, 55] between sample representations in a set. Others compute linear [75, 68] or orthogonal projections [5] from one set of representations to another. These metrics are designed based on a set of desired invariant properties such as permutation invariance or invariance to linear transformations.

Such approaches have been commonly applied in neuroscience for measuring representational distance of activations from networks and activity in the brain to understand which neural networks are architecturally most similar to the brain [75, 14, 71, 24]. Under this context, Han et al. [30] has shown the inability of current representational distance metrics – specifically the metric used here, centered kernel alignment – to distinguish representations based on architecture. This paper provides the foundation for our intuition that networks may have similar representations that allow for transferring inductive biases from one network to another.

**Untrainable Networks**: This work examines RNNs and transformers for sequence modeling and FCNs and plain CNNs for image classification. Prior work explored similar approaches but performed poorly compared to the guide networks we leverage for improved training.

In sequence modeling, classical RNNs [69, 58, 13, 29] were constrained by vanishing and exploding gradients [35], making them unsuitable for long sequence tasks requiring memorization [27]. Gradient flow techniques were developed, but significant progress came from architectures like LSTMs [34] and transformers [74]. Transformers, however, have been found untrainable on formal language tasks requiring full-sequence reasoning, where RNNs succeed [8].

For image classification, small feed-forward networks with 3-5 hidden layers and fewer than 100 units per layer were trained on object recognition datasets [53, 6, 41, 57]. These efforts prioritized training fit over generalization performance and achieved low results [53, 6]. Strategies to reduce overfitting, such as topological structure [67] or early stopping [9], were hindered by complex designs and hyperparameter tuning, leading to poor training fits. Further methods used alignment between thin deep FCNs and wide shallow FCNs to prevent overfitting, an approach similar to our paper [64]. Deep convolutional networks were also applied to image classification [47, 7] but struggled with vanishing gradients, limiting their depth.

**Model Distillation**: Guidance shares a resemblance with model distillation [33, 26, 66, 36]. Distillation transfers knowledge from a teacher model to a student model by introducing a new component to the loss function that enforces the student model to behave like the teacher model [42, 76]. This usually consists of penalizing the KL-divergence between the logit predictions of the student and teacher model.

Representation-based distillation [72, 10, 50] and alignment techniques have been proposed to improve alignment between two networks. Certain works have proposed correlation congruence or similarity preserving metrics [62] for aligning two networks, particularly as a way to do architecture search between CNNs [4]. Methods have been proposed that use CKA as an alignment approach between representations of two networks or with representations in the brain with notable improvement in network performance [65, 18].

We distinguish guidance from distillation. Guidance can use a smaller untrained guide instead of a larger trained teacher. This is due to guidance operating over intermediate activations of the network instead the output of the network probabilities or output features, like distillation does. Guidance also operates at many levels at the same time, aligning many layers at once. This helps address the credit assignment problem that gradient descent has when tuning weights early in a network. We also consider many more networks for guidance than is traditional for distillation including networks which have very different architectures like Transformers to RNNs. Distillation is usually carried out between two closely related architectures. We apply guidance to do the opposite.

## 3 Methods

Guidance introduces a term in the loss of a target network, $\mathcal{N}^T$, to encourage representational alignment with a guide network, $\mathcal{N}^G$. We update only the target's parameters, $\theta^T$, while keeping the guide's parameters, $\theta^G$, frozen. On each minibatch, we compute a similarity metric $\mathcal{M}$ (e.g., CKA) between guide-layer activations $\mathbf{A}_{i^G}^G$ and matched target activations $\mathbf{A}_{i^T}^T$. We refer to the correspondence between layers of the guide $\{i^G\}$ and layers of the target $\{i^T\}$ as $I$. While this correspondence, $I$, could be complex as any two architectures can form a guide/target pair, here we choose architectures that make the correspondence obvious as is discussed later. For example, the stacked RNNs and Transformers have the same number of layers in our experiments.

The target and guide receive the same input. Per minibatch, we collect activations from intermediate layers of both networks. Layers of guide network are mapped to layers of the target network; see fig. 1. We formulate the loss in terms of minimizing the *representational dissimilarity*, $\bar{\mathcal{M}}$, i.e., the complement of a representational similarity metric, between guide and target activations layer by layer, summing the results. Here we use linear CKA, though any differentiable similarity metric could plug into Eq. (1). Efficiency or incremental computation is much more important than it is in traditional applications since this operation happens for every minibatch. We discuss details in appendix B.1.

Given $\mathcal{L}_T$ as the original loss of the target network, the guide network's original loss function is irrelevant. The guide could be pretrained on another task. In fact, it need not even have been trained at all, only its architecture shapes the target. This latter setting is what allows transferring architectural priors without transferring knowledge from the guide to the target, as there is none in a randomly initialized guide. See eq. (1) for an overall loss. We discuss details in appendix B.

$$\mathcal{L}(\theta^T) = \mathcal{L}_T(\theta^T) + \sum_{i \in I} \bar{\mathcal{M}}(\boldsymbol{A}_{i^T}^T(\theta^T), \boldsymbol{A}_{i^G}^G(\theta^G)) \tag{1}$$

Equation (1) minimizes a task loss while increasing alignment between the target and guide networks given the mapping between them. The mapping may be sparse; not every layer needs to be used. This is important for guidance with transformers or stacked RNNs, as will be explained later. Note that the guide's parameters, $\theta^G$, are constants, i.e., the guide is never updated.

Metrics like CKA can capture and encode inductive biases in neural network computations. For instance, CKA is a measure that depends on second-order statistics, specifically pairwise sample distance matrices. Architectural choices imprint distinct features on those statistics. For example, consider local receptive fields in a convolutional layer. Units that cover neighboring pixels receive correlated input, and this is reflected in our activations. Such correlations are reflected in our distance matrices, and these can be transferred to distance matrices associated with FCN layers that lack local correlations. Similarly, weight sharing, where the same kernel is applied at every spatial location, will also be reflected in a distance matrix.

**Layerwise Mapping**   We design a simple method for mapping guide layers to target layers as part of providing supervision. The goal of this method is to make guide and target networks architecturally agnostic i.e. we can supervise any target network with any guide network.

| Tasks | Guide Networks | Target Networks |
|---|---|---|
| Copy-Paste | Transformer | RNN |
| Parity | RNN | Transformer |
| Language Modeling (Small and Large) | Transformer | RNN |
| Image Classification | ResNet-18 | Deep FCN
Wide FCN |
| | ResNet-50 | Deep ConvNet |

Table 1: **Guide and target networks across tasks**. Our network designs include several untrainable target networks and corresponding trainable guide networks.

As a simple approach, we evenly spread layer computations of our guide network over our target network. For example, if we consider ResNet-18 and a 50-layer FCN, we would map every convolutional ResNet layer to every second or third linear layer of the FCN. Intuitively, evenly spacing guide-to-target matches encourages the target to approximate the guide's compositional functions. Through the design of evenly spreading layers of our ResNet-18, we are guiding the FCN to find a function similar to the guide network.

For our mapping, we consider activations from all tunable-weight layers (convolutional, linear, or RNN/LSTM). For multiple stacked RNNs, LSTMs, or transformers, we extract feature representations from intermediate layers in the stack as well. Using all layers is useful for guidance as it provides a strong signal to induce alignment between the target and guide networks during training. We empirically find that more layers leads to stronger results. Skipping layers based on non-linear transformations reduces memory overhead associated with storing representations per batch.

## 4  Experiments

We design several settings with different target and guide networks to thoroughly test our approach. We include a range of image and sequence modeling tasks. In choosing target networks, we consider a broad range of designs for networks that are not traditionally applied (e.g., a FCN in image classification).

To systematically evaluate our approach, we incorporate two settings. (1) **Untrainable Architectures**: Experiments where the target networks are difficult to train due to architectural limitations, irrespective of the task. For example, memory incorporation in RNNs or overfitting in deep FCNs. (2) **Untrainable Tasks**: Experiments where certain tasks are inherently challenging for specific architectures, making them untrainable without additional supervision. For example, sequence classification with transformers.

**Tasks**: We describe the task settings. We consider three sequence modeling tasks to allow for a broader range of architectural settings. We first consider a task called *copy-paste* [27]. In this task, we generate a sequence of numbers in the range of $1 - 10$. The model is trained to recover the same sequence in the output. In our setting, we consider sequence lengths that range from 20 to 40 values total (internal sequence and padding). We generate a copy-paste dataset, sampling sequences containing numbers between 1 and 10. We generate a total of $100,000$ examples, training on $80,000$ examples, validating on $10,000$ examples, and testing on $10,000$ examples.

We also include the *parity* task, a binary classification task where a model is fed a bitstring and outputs 1 when there is an even number of ones in the bitstring and 0 otherwise. We generate a series of bitstrings with sequence lengths that range from 2 to 50 as done in prior work [8].

Finally, we consider a *language modeling* task using the WikiText-103 dataset [54] where models must predict the next token given some context. This uses the train, validation and testing splits defined by the WikiText dataset and for all experiments, we use a context length of 50. We tokenize the text data using the GPT-2 [61] tokenizer.

For an image-based task, we focus on *image classification* and use the ImageNet-1K dataset [19] for training and testing. We use the splits defined by the dataset. We report accuracy on the validation set for all experiments.

**Architectures**: For all tasks, we describe our target untrainable architectures for each task separately as well as the guide networks that are employed to make the untrainable network trainable. We give an overview in table 1. We provide further details in appendix C.

*Sequence Modeling*: For our copy-paste task, we use a vanilla, 4-layer RNN as our target network. In copy-paste, architectural and algorithmic limitations make RNNs an untrainable architecture. For our language modeling task, we include two settings with a small (4 layer) and large (6 layer, larger hidden dimension) RNN. In this setting, vanishing gradients and limited context incorporation make RNNs an untrainable architecture as the training loss saturates. For the parity task, we use a 1-layer transformer encoder architecture, similar to prior work [8, 28]. For the copy-paste task, we train a guide network, 4-layer transformer decoder model which achieves 96.90% accuracy. Similarly, for language modeling, we train a 4-layer transformer decoder guide network with a context window of 256. Our final perplexity is 34.15 for the small language modeling setting and 33.10 for the large language modeling setting. For the parity task, we train a 1-layer vanilla RNN as a guide network which achieves 100% accuracy as reported by [8].

*Image Classification*: We use three target networks: Deep FCN, Wide FCN, and Deep ConvNet. Deep FCN is a fully-connected network with 50 blocks consisting of feedforward layers followed by non-linearities. This network is an untrainable architecture, lacking inductive biases to prevent overfitting and having vanishing gradients. Wide FCN is a fully connected network with 3 blocks with feedforward layers that have 8192 units. This is categorized as an untrainable task due to a saturation in the training performance. Deep ConvNet is the same architecture as ResNet-50 [31], but without residual connections. This is categorized as an untrainable architecture due to the vanishing gradient problem. We use two guide networks: ResNet-18 and ResNet-50. ResNet-18/50 is a deep convolutional network with 18/50 convolutional blocks and residual connections. We refer to He et al. [31]. We supervise the Deep FCN and Wide FCN with ResNet-18 and supervise the Deep ConvNet with ResNet-50.

**Training**: For each setting, we train the base target network and perform an experiment where both a trained and untrained guide network supervises the base target network. All networks are trained with cross-entropy loss, without loss of generality. For all sequence modeling tasks, i.e. copy-paste, parity, and language modeling we use AdamW [52]. For language modeling, we also incorporate gradient clipping due to unstable training with long sequences. When training networks for image classification using ImageNet-1K, we use the Adam [43] optimizer.

To ensure consistency of comparisons across learning curves, we use a consistent batch size of 256. Representational similarity metrics are affected by the number of samples in the calculation, where more samples allows for the metric to approximate representational distance better. We use 256 as a proxy, dependent on GPU memory, although more memory would allow for bigger batch sizes with potentially better results. Due to the large number of training settings, we employ several different learning rates. We tune the learning rate carefully for baseline training to ensure maximal performance. We sweep the parameter across 5 different values and choose the results with the lowest validation loss. This ensures we are choosing the training with the best performance.

After choosing the optimal learning rate, we then train all networks and settings for 100 epochs with 5 random seeds to compute error bars. Our error bars are associated with the standard error across each step across all seeds. We choose the seed-based average test accuracy associated with the epoch with the lowest seed-based average validation loss.

## 5  Results

**Sequence Modeling**: On the copy-paste task, guiding a 4-layer RNN with a Transformer improves copy-paste accuracy by over 25%. See fig. 2 and table 2 Previous studies blamed RNN failures on vanishing gradients and memorization limits. Our results show a potential optimization scheme for RNNs that is applicable for sequence memorization. Remarkably, a random (untrained) Transformer guide outperforms a trained one, suggesting pure architectural bias drives gains. We believe this is because optimization with randomly initialized networks is easier due to the degrees of freedom in CKA. See appendix E. We plan to explore this more thoroughly in further analyses. These gains persist under our layerwise ablations and metrics; see appendix K and appendix G.

On the parity task, a 1-layer Transformer guided by an RNN improves its test accuracy by 7%. This is a complementary result to copy-paste and language modeling where the guide network was a

| Experiment | Copy-Paste Accuracy ($\uparrow$) | Parity Accuracy ($\uparrow$) | Language Modeling (Small) Perplexity($\downarrow$) | Language Modeling (Large) Perplexity ($\downarrow$) |
|---|---|---|---|---|
| RNN | $14.35 \pm 0.01$ | 100 | $69.19 \pm 1.89$ | $89.13 \pm 2.00$ |
| Untrained RNN | — | $2.32 \pm 0.41$ | — | — |
| Transformer | 96.98 | $71.98 \pm 3.16$ | 34.15 | 33.10 |
| Untrained Transformer | $1.04 \pm 0.81$ | — | $5.19\mathrm{e}5 \pm 90.44$ | $5.19\mathrm{e}5 \pm 90.44$ |
| RNN $\rightarrow$ Transformer | — | $\mathbf{78.49} \pm 2.16$ | — | — |
| Untrained RNN $\rightarrow$ Transformer | — | $70.38 \pm 4.17$ | — | — |
| Transformer $\rightarrow$ RNN | $23.27 \pm 1.02$ | — | $\mathbf{40.01} \pm 1.54$ | $\mathbf{36.91} \pm 1.04$ |
| Untrained Transformer $\rightarrow$ RNN | $\mathbf{42.56} \pm 1.51$ | — | $59.61 \pm 2.33$ | $47.17 \pm 2.50$ |

Table 2: **Guidance improves performance for sequence modeling**. RNN performance improves dramatically when aligning with the representations of a Transformer for copy and paste, as well as for language modeling with small and large RNN architectures. RNNs close most of the gap to Transformers for language modeling and are likely competitive with further scale. Transformers in turn, improve parity performance when aligning with an RNN. Guidance is able to transfer priors between networks.

| Experiment | ImageNet Top-5 Validation Accuracy ($\uparrow$) |
|---|---|
| ResNet-18 | 89.24 |
| Untrained ResNet-18 | $0.24 \pm 0.043$ |
| ResNet-50 | 92.99 |
| Untrained ResNet-50 | $0.54 \pm 0.029$ |
| Deep FCN | $1.65 \pm 1.21$ |
| ResNet-18 $\rightarrow$ Deep FCN | $7.50 \pm 1.51$ |
| Untrained ResNet-18 $\rightarrow$ Deep FCN | $\mathbf{13.10} \pm 0.72$ |
| Wide FCN | $34.09 \pm 0.91$ |
| ResNet-18 $\rightarrow$ Wide FCN | $\mathbf{43.01} \pm 0.92$ |
| Untrained ResNet-18 $\rightarrow$ Wide FCN | $39.47 \pm 0.31$ |
| Deep ConvNet | $70.02 \pm 1.52$ |
| ResNet-50 $\rightarrow$ Deep ConvNet | $\mathbf{78.91} \pm 2.16$ |
| Untrained ResNet-50 $\rightarrow$ Deep ConvNet | $68.17 \pm 2.54$ |

Table 3: **Guidance improves performance for image classification**. Alignment with a ResNet dramatically improves a deep FCN, particularly with an untrained ResNet. Significant gains are seen with a wide FCN as well. Deep CNNs without residuals gain only with a trained ResNet. Across all settings, guidance can help train architectures that were otherwise considered unsuitable.

transformer and our target network was a RNN. This improves over results from several prior papers [8] that have pointed out fundamental limitations of transformers to perform formal language tasks.

Unlike copy-paste, the performance improves when using a trained RNN as the guide network. This could be due to the wide gap in performance between an untrained RNN and trained RNN on parity. Parity uniquely benefits from learned positional encodings in the trained RNN, which the Transformer lacks. This information is likely crucial to the transformer, which has limited sequence pooling capacity and fewer degrees of freedom.

On language modeling, similar to copy-paste, guided RNNs halve the perplexity from ~70 points to 35 points on WikiText-103, closing in on Transformer baselines. While performance generally saturates for the 4-layer RNN, guidance continuously improves the RNN performance by over 30 points for text perplexity for both trained and randomly initialized guide networks. Scaling up to a 6-layer RNN further cuts perplexity by 10 points, indicating guidance scales with model size. This implies that information from the transformer can be transferred to the RNN. We also believe that this has exciting implications for scaling laws with RNNs. We see a similar trend with a randomly initialized transformer as the guide network, implying that architectural priors in the transformer are driving improvement in guided network performance.

**Image Classification**: Guidance boosts validation accuracy by 5–10% across our Deep FCN, Wide FCN, and Deep ConvNet; see table 3. We also observe significantly better loss curves from a better fit with the training loss and reduced overfitting with the validation loss. An untrained ResNet-18 guide

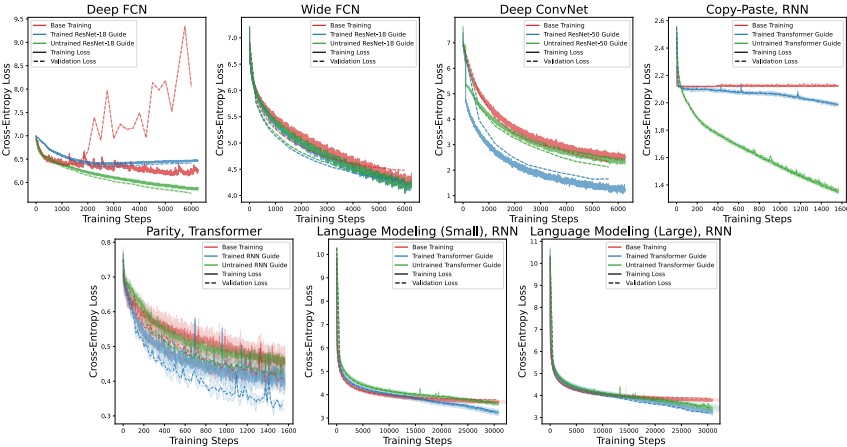

Figure 2: **Training and validation under guidance for all experiments reported in table 1**. For every result in Table 3 and Table 2, we show the training and validation loss with error bars across multiple runs, although these are often too small to see. Note that often the best results occur with the untrained guide.

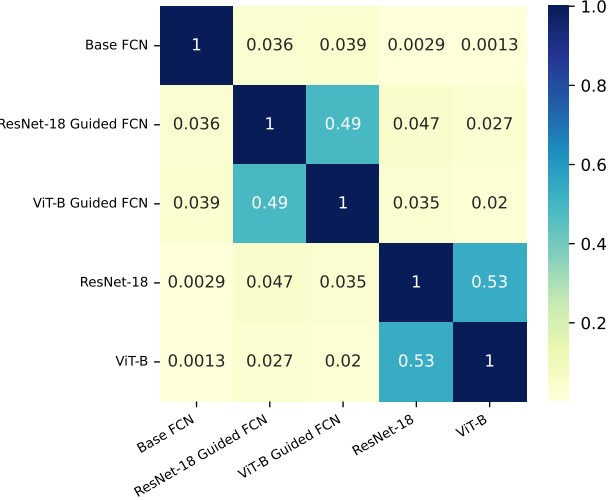

Figure 3: **Guidance aligns error consistency**. The relationship between the guide networks is mirrored in that of the guided networks, even when the target is entirely unlike the guides initially. This is additional evidence that guidance doesn't just improve performance arbitrarily; the target becomes more like the guide.

outperforms its trained counterpart on Deep FCN, underscoring pure architectural priors like with copy-paste. For example, the Deep FCN results in the top left of fig. 2 are significantly better with a randomly initialized ResNet-18 as the guide network instead of a trained ResNet-18. This trend also occurs with Wide FCN. We show results with other neural distance functions in appendix J.

The Deep ConvNet (no skips) gains only from a trained ResNet-50 guide—implying residual connections require learned weights to shape representation. This explanation provides an additional interpretation for the role of residual connections and their influence on the representation space. This indicates that residual connections must be trained to have an influence on the representation space. This aligns with prior studies of residual connections [39, 32].

**Error Consistency**: Guided FCNs mirror the ResNet–ViT error overlap, proving they inherit the guide's decision patterns. Using Deep FCN as our target model, we guide it with a ResNet-18 or a ViT-B-16 [21]. We then measure the error consistency [23] between all of the networks; see fig. 3. The error consistency between the initial FCNs is entirely unlike the ResNet-18 or ViT-B. Guidance creates two FCNs which have the same relationship to one another. The ResNet-18-guided FCN and ViT-B-guided FCN have the same error consistency with respect to one another as ResNet-18

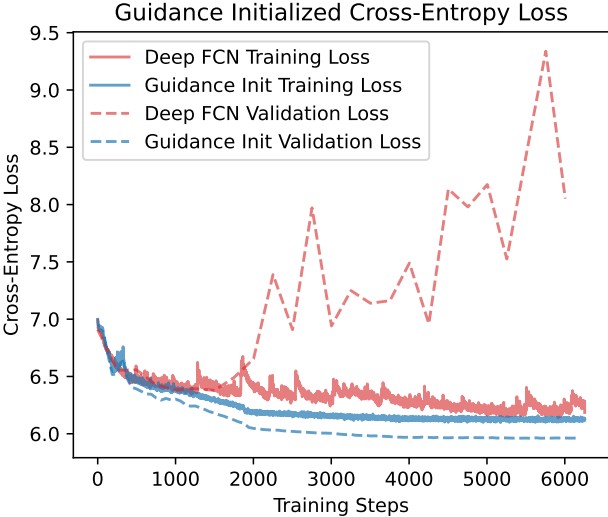

Figure 4: **Initializing fully connected networks with guidance can overcome overfitting.** First, we align a Deep FCN to a random ResNet-18 on noise for 300 steps, then train normally. This two-stage scheme mirrors full guidance, and leads to a similar performance improvement. This suggests that FCNs have guidance-inspired initializations that avoid overfitting.

and ViT-B do. It's not just that the FCN gets generically better; it adopts a prior from the original architecture. We provide further details of the error consistency metric in appendix F.

**Network Initialization**: Is guidance needed throughout training, or is the effect of the guide to move the target into a regime where the two are aligned and the target can be optimized further without reference to the guide? The answer to this question can shed light on whether the guide is telling us that better initializations are likely to exist for the target. To answer this question, we minimize the representational dissimilarity between our target and guide network for a nominal number of training steps, 300. Then we apply task training on the resulting target network with no guidance. Pre-aligning FCN layers to a random guide for 300 steps stops overfitting entirely—no ongoing guidance needed; see fig. 4. Furthermore, while preventing overfitting, we have lower training loss from guidance, indicating a better fit. This implies that there exists a better initialization for FCNs.

**Guidance vs Distillation**: We include a comparison between guidance and knowledge distillation [33] in fig. 5. We find that guidance improves significantly over distillation, particularly when the teacher network is untrained. This is significant, demonstrating that guidance can exploit untrained networks for transferring inductive biases. This indicates that matching internal representations provides a much stronger signal over just matching output behavior. We discuss further in appendix H.

## 6 Conclusion

We demonstrated that guidance eliminates the failure modes of networks previously considered unsuitable or ineffective for specific tasks. Aligning with another network overcomes these shortcomings by transferring inductive biases—either architectural and knowledge-based, or solely architectural when using an untrained guide. This allows guidance to distinguish tasks and architectures that are dependent on architectural biases rather than learned biases. We provide further explanations and intuition for guidance in appendix I. There are many potential aspects that may distinguish between the effects of architectural and inductive biases in improving performance of these architectures, which we aim to explore in future work.

This also opens the door to many applications. Our method can be used to study representational and functional design of neural networks in new ways to reanalyze prior theory of neural network optimization. For example, we can understand distances between architectural components based on which target networks are easier to guide with a particular guide network. We also refine this notion to include a narrow channel through which guidance can occur, the representational similarity. This can serve as a kind of probe.

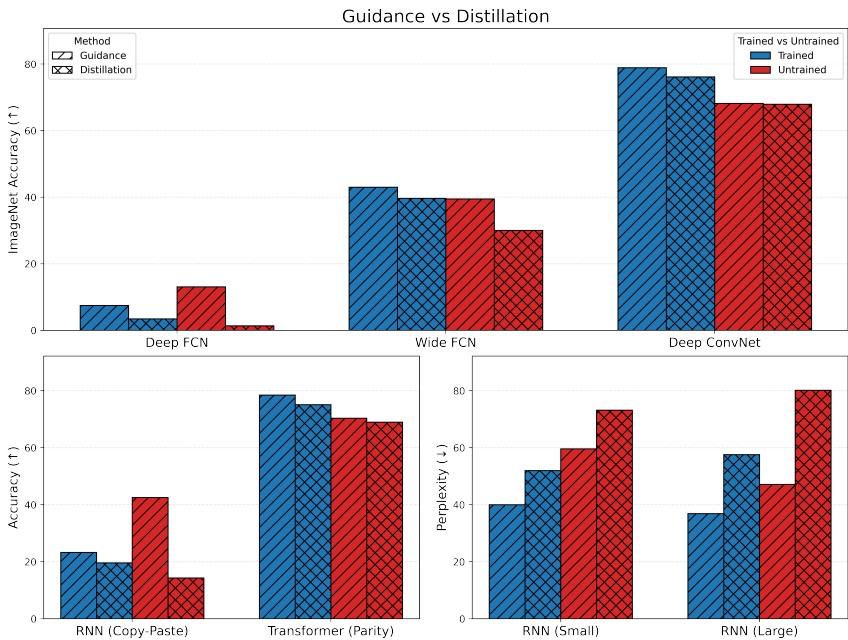

Figure 5: **Guidance outperforms distillation**: We include a comparison between guidance and distillation for all settings with trained and untrained guide networks/teacher networks. We find that guidance outperforms distillation in all settings, highlighting that, unlike guidance, distillation fails in settings with an untrained teacher.

We believe guidance also reveals a new conceptual lens on inductive biases and architectures. When we use an untrained guide network, we reveal what an architecture by itself brings to the table. Guidance with a trained guide reveals how much learned representations can change performance. These are different phenomena whose root causes are not understood at present but that could elucidate the relationship between priors and architectures.

Our results suggest practical applications by significantly narrowing the performance gap between vanilla stacked RNNs and Transformers and showing signs of scaling, albeit in small-scale experiments with 150M parameters or fewer. Given that stacked RNNs are equivalent to single-layer RNNs, most directly to delayed RNNs [73], this implies that complex modifications to RNNs may be unnecessary for language modeling. For other networks, like fully connected ones, we only overcame the initial obstacle. Further research is needed to refine these into effective vision models, as they avoid immediate overfitting. In the future, we hope to find methods for making these networks competitive with their guide networks.

Guidance also proved to be a tool with which to discover the possibility of new initializations. At the moment, no known method exists to find better initializations for networks. In some cases as with the FCNs for vision, guidance can be disconnected after a nominal number of steps, but still goes on to regularize the target network. This strongly implies that an initialization regime for that target with the same regularization exists. This is all that guidance could do in that case. We now need tools to go backwards, given networks which are correctly initialized and networks which are not, discover what that initialization is. This is a much better place to be in. A systematic sweep of targets and guides to look for better initializations should be carried out.

Looking into the long-term future, guidance invites us to treat architecture itself as a trainable prior added directly to a generic network's loss. Because guidance can rescue models that previously overfit or underfit, we can revisit designs abandoned during neural architecture search. These threads demonstrate the major possibility for future work in this space. Guidance is a tool, not a finished and well-understood theory or doctrine. Tools are useful for enabling more discoveries. We believe that these discoveries will allow the community to more easily pursue questions that training difficulties might have prevented in the past, especially with regard to understanding the relationships between architectures.

## Acknowledgements

This work was supported by the Center for Brains, Minds, and Machines, NSF STC award CCF-1231216, the Brains, Minds, and Machines Summer School, the NSF award 2124052, the MIT CSAIL Machine Learning Applications Initiative, the MIT-IBM Watson AI Lab, the CBMM-Siemens Graduate Fellowship, the DARPA Mathematics for the DIscovery of ALgorithms and Architectures (DIAL) program, the DARPA Knowledge Management at Scale and Speed (KMASS) program, the DARPA Machine Common Sense (MCS) program, the Department of the Air Force Artificial Intelligence Accelerator under Cooperative Agreement Number FA8750-19-2-1000, and the Air Force Office of Scientific Research (AFOSR) under award number FA9550-21-1-0399. The views and conclusions contained in this document are those of the authors and should not be interpreted as representing the official policies, either expressed or implied, of the Department of the Air Force or the U.S. Government. The U.S. Government is authorized to reproduce and distribute reprints for Government purposes notwithstanding any copyright notation herein. V.S. and D.M. are supported by the National Science Foundation Graduate Research Fellowship under Grant No. 2141064. Any opinion, findings, and conclusions or recommendations expressed in this material are those of the authors and do not necessarily reflect the views of the National Science Foundation.

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

# A    Appendix Overview

We present additional details of guidance, experiments and analysis, as well as additional results. In Appendix B, we provide additional details for our guidance approach, with a full explanation of centered kernel alignment. In Appendix C, we provide additional details on our architectural designs and network training. In Appendix D, we introduce a new experiment where we feed noise to our guide network rather than real data and compute representational alignment, leading to similarly improved results. This further establishes a transfer of a prior rather than knowledge. In Appendix E, we show visualizations of the representational dissimilarity loss over training to give context of dynamics over training and show additional explanations for results with trained and randomly initialized guides. In Appendix F, we provide further explanation of error consistency as a measure of functional similarity between networks. In Appendix G, we provide test accuracy metrics over training as a complementary of network performance over training outside of cross-entropy loss. In Appendix H, we provide a comparative baseline to guidance, distillation [33]. We find that basic distillation performs worse in comparison. In Appendix I, we provide an interpretation and analysis of guidance to better characterize and understand results based on changes in the internal geometry of a network before and after guidance. In Appendix J, we apply guidance with additional neural distance functions including RSA [46] and ridge regression. We find a correlation between the success of our approach and degrees of freedom of a particular distance function. In Appendix K, we provide additional ablations over guidance such as guiding a certain number of layers or guiding specific layers of a network.

# B    Methods Overview

We give an overview of guidance in algorithm 1 and highlight crucial changes to base neural network training in either red or blue. We use blue to indicate the collection of network activations and red to indicate the layerwise mapping and representational alignment using a distance metric. This gives an overview of our layer mapping between the target and guide network. Crucially, we find that the simplest layer mapping where we evenly distribute guide network layers across target network layers for supervision obtains strong results.

---

**Algorithm 1 Guidance**: Guide Network Representational Alignment

---

**Require:**  Target network; $\mathcal{N}^T$ with parameters $\theta^T$; Guide network $\mathcal{N}^G$; Dataset $\mathcal{D} = \{(x_i, y_i)\}_{j=1}^{N}$; Representational Distance Metric $\bar{\mathcal{M}}$; Loss function $\mathcal{L}^T$

1:  **for** $j = 1 \rightarrow N$ **do**
2:        # Base training with vanilla loss function
3:        outputs $\leftarrow \mathcal{N}^T(x_j)$
4:        loss $\leftarrow \mathcal{L}_T(\text{outputs}, y_j \mid \theta^T)$
5:        # collect layer activations
6:        $\{\boldsymbol{A}_{iT}^T\}_{iT=1}^{t} \leftarrow \text{activations}(\mathcal{N}^T(x_j))$
7:        $\{\boldsymbol{A}_{iG}^G\}_{iG=1}^{l} \leftarrow \text{activations}(\mathcal{N}^G(x_j))$
8:        # Get step size between the number of layers between the two networks for layer mapping.
9:        **if** $l > 1$ **then**
10:            $step \leftarrow (t-1)/(l-1)$
11:        **else**
12:            $step \leftarrow 1$
13:        **end if**
14:        # Map the layers and add up layer-wise representational distance
15:        total $\leftarrow 0$
16:        **for** $i = 1 \rightarrow l$ **do**
17:            index $\leftarrow \min(\text{round}(i \times step), t-1)$
18:            rep $\leftarrow \mathcal{M}(\boldsymbol{A}_{\text{index}}^T, \boldsymbol{A}_{\text{index}}^G)$
19:            total $\leftarrow$ total + rep
20:        **end for**
21:        loss $\leftarrow$ loss + total
22:  **end for**

---

### B.1 Centered Kernel Alignment

To compare representations, we use a representation similarity metric, $\mathcal{M}$, which corresponds to centered kernel alignment (CKA) [45, 16, 17] in our setting. We specifically consider linear CKA.

CKA uses kernel functions on mean-centered representations to compute representational similarity matrices, which are then compared via the Hilbert-Schmidt Independence Criterion (HSIC). More specifically, suppose we have two sets of representations $\boldsymbol{R} \in \mathbb{R}^{b \times d_1}$ and $\boldsymbol{R}' \in \mathbb{R}^{b \times d_2}$. We first compute the Gram matrices for each set of representations

$$\boldsymbol{K} = \boldsymbol{R}\boldsymbol{R}^T, \boldsymbol{L} = \boldsymbol{R}'\boldsymbol{R}'^T \tag{2}$$

We center the Gram matrices by introducing a matrix, $H$, where $H = \boldsymbol{I}_b - \frac{1}{n}\boldsymbol{1}\boldsymbol{1}^T$.

$$\tilde{\boldsymbol{K}} = \boldsymbol{H}\boldsymbol{K}\boldsymbol{H}, \tilde{\boldsymbol{L}} = \boldsymbol{H}\boldsymbol{L}\boldsymbol{H} \tag{3}$$

We compute the HSIC on the Gram matrices.

$$HSIC(\boldsymbol{K}, \boldsymbol{L}) = \text{tr}(\tilde{\boldsymbol{K}}, \tilde{\boldsymbol{L}}) \tag{4}$$

Finally, we define our linear CKA metric as:

$$\mathcal{M}(\boldsymbol{R}, \boldsymbol{R}') := \text{CKA}(\boldsymbol{K}, \boldsymbol{L}) = \frac{HSIC(\boldsymbol{K}, \boldsymbol{L})}{\sqrt{HSIC(\boldsymbol{K}, \boldsymbol{K}) * HSIC(\boldsymbol{L}, \boldsymbol{L})}} \tag{5}$$

In our setting, we consider representational *dissimilarity* and aim to minimize the dissimilarity between representations from our target network and guide network. We define this as:

$$\bar{\mathcal{M}}(\boldsymbol{R}, \boldsymbol{R}') = 1 - \mathcal{M}(\boldsymbol{R}, \boldsymbol{R}') \tag{6}$$

Linear CKA ranges from 0 (identical representations) to 1 (very different representations). Because of this, we take the complement by subtracting the linear CKA from 1 to represent dissimilarity.

### B.2 Methodology Limitations

Our guide network supervision through representational alignment has one primary limitation due to increased memory usage during training. Due to saving activations across several layers of the two networks, GPU memory usage increases dramatically. Moreover, our methodology works better as batch size increases since this allows for better approximation of representational similarity, increasing memory usage even more. Furthermore, including more layers for supervision leads to improved results.

In this paper, we introduce simple techniques to handle memory constraints such as gradient accumulation and gradient checkpointing [51]. In practice, more memory optimization techniques may become necessary to consider larger untrainable networks. Further work could consider using stronger representational alignment strategies to reduce the number of samples necessary to achieve a strong fit.

## C   Architecture and Training Details

### C.1   Architectural Design Details

For all tasks, we describe our target untrainable architectural designs for each task separately as well as the guide networks that are employed to make the untrainable network trainable.

### C.1.1 Copy-Paste

**Target Networks**

*RNN*: We design a 4-layer RNN with a hidden dimension of 768 units, followed by a fully connected layer. In copy-paste, architectural and algorithmic limitations make RNNs an untrainable architecture for the task. Specifically, RNNs must memorize the input sequence which is difficult, particularly with a padding token. RNNs are generally considered to be inapplicable to the copy-paste task.

**Guide Networks**

*Transformer*: We consider a 4 layer transformer decoder architecture with a hidden dimension of 768 units across 12 transformer heads. The transformer is well-suited for copy-paste as the attention mechanism can act as a routing mechanism for the sequence. We train the transformer guide from scratch, as with language modeling and achieve $96.90\%$ accuracy on the task.

### C.1.2 Parity

**Target Networks**

*Transformer*: Similar to Bhattamishra et al. [8], we design a 1 layer transformer encoder network with a hidden dimension of 64 units across 4 attention heads. Transformers have lower accuracy on formal language tasks that require reasoning over a sequence in comparison to traditional sequence models [28]. Due to the enormous gap in performance and saturation of performance, we categorize the transformer as untrainable.

**Guide Networks**

*RNN*: We include a 1 layer vanilla RNN with a hidden dimension of 64 units. Similar to Bhattamishra et al. [8], we achieve 100% accuracy on the task.

### C.1.3 Language Modeling

We include two language model settings to test scaling in RNNs. The first setting uses small networks, which we refer to as Small RNN and small Transformer. The other uses a large RNN and large Transformer.

**Target Networks**

*Small RNN*: We design a 4 layer vanilla RNN with a hidden dimension of 512 and with a ReLU activation function. We train this on sequences with a context length of 75. This makes the network untrainable due to problems associated with exploding and vanishing gradients during backpropagation through time.

*Large RNN*: We design a 6 layer vanilla RNN with a hidden dimension of 1024 and with a ReLU activation function. We train this on sequences with a context length of 128. Prior work has demonstrated that larger RNNs are difficult to train in practice [49].

**Guide Networks**:

*Small Transformer*: We design a 4 layer transformer decoder network with 16 attention heads and a hidden dimension of 512. We train the transformer on WikiText-103 with a context length of 256 and achieve a final test perplexity of 34.15.

*Large Transformer*: We design a 4 layer transformer decoder network with 16 attention heads and a hidden dimension of 1024. We train the transformer on WikiText with a context length of 256 and achieve a final test perplexity of 33.10.

### C.1.4 Image Classification

**Target Networks**

*Deep FCN*: We design a fully-connected network consisting of 50 blocks. Each block contains a feedforward linear layer, a batch normalization, and a ReLU nonlinear activation. The intermediate feedforward linear layers contain 2048 units. This network is untrainable due to vanishing gradients since the network is very deep and due to overfitting.

| Tasks | Experiment | Learning Rate |
|-------|------------|---------------|
| Copy-Paste | RNN | $1 \times 10^{-4}$ |
| | Transformer | $1 \times 10^{-4}$ |
| | Transformer $\rightarrow$ RNN | $1 \times 10^{-4}$ |
| | Untrained Transformer $\rightarrow$ RNN | $1 \times 10^{-4}$ |
| Parity | Transformer | $1 \times 10^{-3}$ |
| | RNN | $1 \times 10^{-2}$ |
| | RNN $\rightarrow$ Transformer | $1 \times 10^{-3}$ |
| | Untrained RNN $\rightarrow$ Transformer | $1 \times 10^{-3}$ |
| Language Modeling | Small RNN | $1 \times 10^{-4}$ |
| | Small Transformer | $1 \times 10^{-4}$ |
| | Small Transformer $\rightarrow$ Small RNN | $1 \times 10^{-4}$ |
| | Untrained Small Transformer $\rightarrow$ Small RNN | $1 \times 10^{-4}$ |
| | Large RNN | $1 \times 10^{-4}$ |
| | Large Transformer | $1 \times 10^{-4}$ |
| | Large Transformer $\rightarrow$ Large RNN | $1 \times 10^{-4}$ |
| | Untrained Large Transformer $\rightarrow$ Large RNN | $1 \times 10^{-4}$ |
| Image Classification | Deep FCN | $1 \times 10^{-4}$ |
| | Wide FCN | $1 \times 10^{-4}$ |
| | Deep ConvNet | $1 \times 10^{-3}$ |
| | ResNet-18 $\rightarrow$ Deep FCN | $5 \times 10^{-5}$ |
| | Untrained ResNet-18 $\rightarrow$ Deep FCN | $5 \times 10^{-5}$ |
| | ResNet-18 $\rightarrow$ Wide FCN | $1 \times 10^{-4}$ |
| | Untrained ResNet-18 $\rightarrow$ Wide FCN | $1 \times 10^{-4}$ |
| | ResNet-50 $\rightarrow$ Deep ConvNet | $1 \times 10^{-3}$ |
| | Untrained ResNet-50 $\rightarrow$ Deep ConvNet | $1 \times 10^{-3}$ |

Table 4: **Learning rates for network training**. For all networks, we sweep over 5 learning rate values before choosing the learning rate with the lowest validation loss for training. Our training does not use any learning rate scheduling such as a warm-up scheduler although such techniques may improve results.

*Wide FCN*: We design a network similar to Deep FCN but only containing 3 blocks where each feedforward linear layer contains 8192 units. This network is considered untrainable due to a saturation on the training performance.

*Deep ConvNet*: We design a deep convolutional network with the same architecture as ResNet-50 (convolutional layers followed by batch normalization) but remove the residual connections. This makes the network untrainable due to the vanishing gradient problem as observed in He et al. [31], causing saturation of the loss.

**Guide Networks**

*ResNet-18/50*: A deep convolutional network with 18/50 convolutional blocks and residual connections. We refer to He et al. [31].

We supervise the Deep FCN and Wide FCN with ResNet-18 and supervise the Deep ConvNet with ResNet-50.

## C.2 Training

In Table 4, we show the different learning rate settings we converged to in each experiment. For each experiment, we did a grid search over 5 different learning rate parameters to ensure optimal learning rate setting. We did careful tuning of all training of target networks to ensure maximum performance.

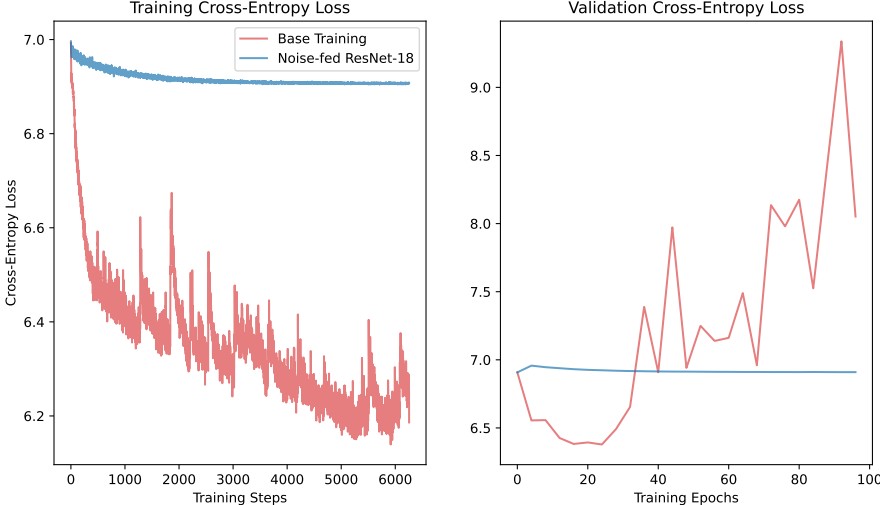

Figure 6: **Feeding noise prevents overfitting**. We introduce an additional experiment where we feed noise to our guide network rather than the same batch during each training step with guide network guidance. We sample noise from a Gaussian distribution with a mean of 0 and a standard deviation of 1. We find that despite having no information about the images in the batch, the guide network still provides an inductive bias to prevent overfitting. While this noise increase the training loss, this shows a true transfer of an inductive bias that is not driven by pure distillation of similar features.

For all image classification tasks, we used the Adam optimizer [43], in-line with prior work [31]. For all sequence modeling tasks, we use AdamW [52], which has been useful in training sequence models like RNNs and Transformers [61].

The training experiments in this paper were completed across 4 H100s and 4 A100 GPUs for 3 weeks in total. GPU optimization techniques were taken such as gradient accumulation and gradient checkpointing and some language modeling experiments used mixed-precision training.

## D   Representational Regularization: Guidance with Noise

We also aim to understand the role of the guide network as in guidance. In all experiments, we use trained and untrained guide networks and see consistent improvements for training the target network. The success of untrained networks implies that our training method is not performing distillation but instead truly transferring a prior from the guide network to the target network. To more strictly test this theory, we include an experiment where we feed noise to the guide network instead of the same batch of data fed to the target network as implied by eq. (1).

We apply this experiment to the Deep FCN with an untrained ResNet-18 as the target network. At each training step, we pass a noisy batch which is sampled from a random Gaussian with mean of 0 and standard deviation of 1. We train for 100 epochs and report the learning curve results in fig. 6.

This result confirms our intuition about the role of guide network: as a guide on model priors rather than a pure distillation of information. While the overall cross-entropy loss magnitudes are higher and the overall accuracy is lower when passing noise to the guide network, our results are significantly better than applying vanilla training approaches to the Deep FCN.

## E   Representational Similarity Loss

We can view the representational alignment between the guide and target networks during training. This allows us to better understand how this representational alignment influences network performance. We show sequence modeling results in fig. 7 and image classification results in fig. 8.

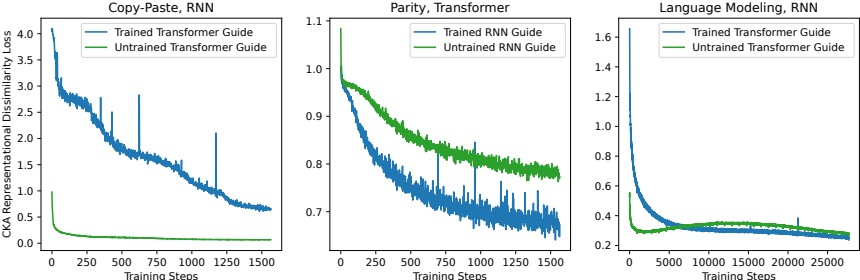

Figure 7: **CKA representational similarity loss for sequence modeling tasks.** We visualize the total CKA dissimilarity loss across all layers across training for all three sequence modeling tasks. The CKA dissimilarity loss represents the representational alignment between our guide network and target network. We can observe that for the copy-paste task and language modeling task, the target network aligns with a randomly initialized network more quickly. This could be because of special properties of RNNs.

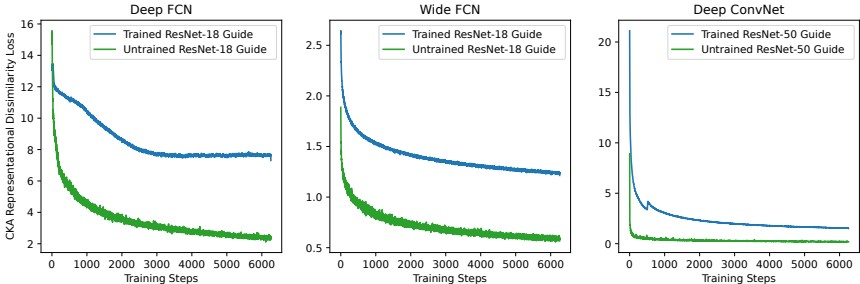

Figure 8: **CKA representational similarity loss for image classification**. We visualize the total CKA dissimilarity loss across all layers across training for image classification. The CKA dissimilarity loss represents the representational alignment between our guide network and target network. We can observe that for Deep FCN and Wide FCN, the target network aligns with a randomly initialized network more quickly. This corresponds with results where randomly initialized guide networks had superior performance to trained guide networks.

We notice that across most tasks, reducing representational dissimilarity is easier with activations from randomly initialized networks rather than trained networks. This provides additional evidence of representational alignment for inductive bias transfer. We notice that for certain cases, such as Parity, the randomly initialized guide network has higher representational dissimilarity loss than the trained guide network. This is matched with the Parity result in table 2 and fig. 2.

However, we can also observe more inconsistent results with the Deep ConvNet where the untrained guide network has lower representational dissimilarity loss than the trained guide network, even at the end of training. One possible explanation that the inductive bias was more similar for Deep ConvNet and ResNet-50. This means that trained features are more important for better Deep ConvNet results and representational alignment with a trained network is important. This result has interesting implications for understanding the role of residual connections. Since untrained ResNet-50 is easier to align with than a trained ResNet-50, this demonstrates that residual connections influence representation spaces during training. The untrained residual connections have little influence on the inductive biases of the network or the overall representation space. This demonstrates the strength of our method as a way to interpret neural network design choices and how they influence representation and functional aspects of a network.

These results are also potentially indicative of architectural properties of RNNs and FCNs which match randomly initialized networks more quickly. For instance, one potential explanation is that RNNs have more degrees of freedom [8] and therefore, only need inductive guidance rather than trained features. Transformers may require learned features indicating that the bottleneck for transformers on the parity is not algorithmic but feature-based. Much of the future work can use these results to design better networks with more informed designs.

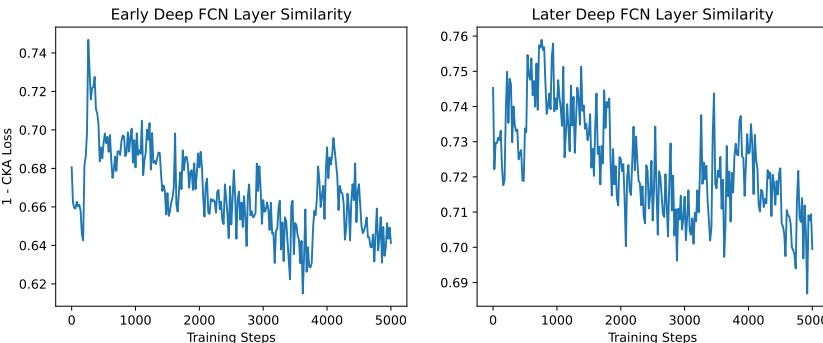

Figure 9: **CKA dissimilarity decreases more in earlier network layers than later layers.** When we separate the CKA dissimilarity across layers of the target network, we find that earlier layers optimize more and earlier. We take two layers from a Deep FCN during guidance with a randomly initialized ResNet-18. The early layer comes from the 15th FCN block. The later layer comes from the 43rd FCN block. We see that both layers are eventually optimized but the later layer receives less supervision and has a higher CKA at the end of training.

## E.1 Layerwise Analysis

We provide a deeper analysis of patterns of the representational dissimilarity across different layers during guidance in fig. 9. We find that earlier layers generally have higher CKA similarities with their corresponding layer from the guide network and later layers have lower CKA similarities. Furthermore, these later layers optimize later in the training process.

## F Error Consistency

We measure *error consistency* ($\kappa$) [23] between the guided target networks which indicates the error overlap between two networks based on the accuracy of the networks, i.e. do the two networks make similar class predictions? The measure first calculates the expected error overlap. Suppose $a_1$ is the accuracy of the first guided network and $a_2$ is the accuracy of the second. The expected error overlap is given by $c_{\exp} = a_1 * a_2 + (1 - a_1) * (1 - a_2)$. Next, we measure the observed error overlap across each sample in the validation set as $c_{\text{obs}} = $ # of samples where both models agree / total trials. Finally, we can write $\kappa$ as:

$$\kappa = \frac{c_{\text{obs}} - c_{\exp}}{1 - c_{\exp}} \tag{7}$$

$\kappa$ ranges from $-1$ to $1$, where $1$ is perfect agreement, $-1$ is perfect disagreement and $0$ is change agreement. When $\kappa > 0$, this implies that models make consistent error patterns, $\kappa < 0$ implies that models make inverse error patterns, and $\kappa \approx 0$ implies independent error patterns.

## F.1 Guide Network Representation Comparison

We contextualize the findings in error consistency by comparing the representations of the guide networks, in this case ResNet-18 and ViT-B-16. We apply a layer mapping between layers of ResNet-18 and ViT-B-16 and compute the representational similarity over 1000 input images. Results are shown in fig. 10.

Our findings are useful for error consistency. If models are inheriting inductive biases from their guide network, then the models would have similar methods to process low-level image features are indicated by a stronger CKA in earlier layers between the ResNet-18 and ViT-B. This means that errors will be consistent for low level features but inconsistent for high level features collected in later layers.

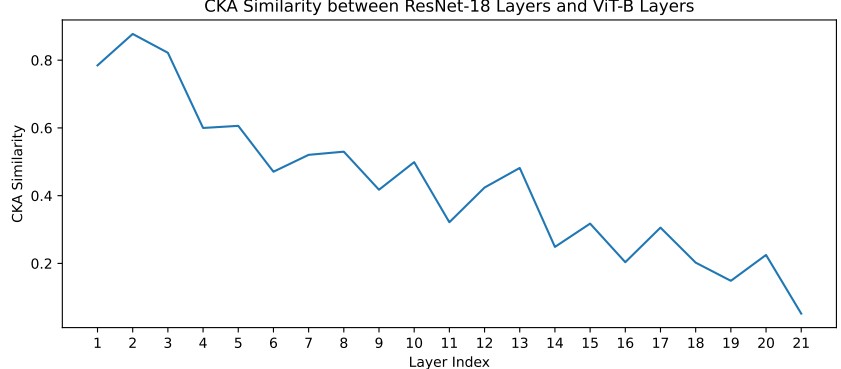

Figure 10: **Earlier layers of ResNet-18 and ViT-B are more similar.** We analyze the representational similarity between activations from layers ResNet-18 and ViT-B-16 via CKA. We find that earlier layers are more similar while later layers have divergent representations. We see that this manifests in distinct error consistency patterns when ResNet-18 and ViT-B are used as guide networks.

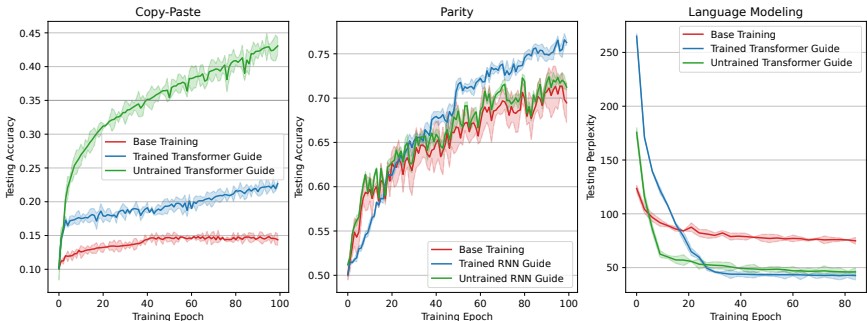

Figure 11: **Testing accuracy improves across guidance for sequence modeling**. We visualize the testing accuracy for sequence modeling as an example to demonstrate that guidance improves accuracy across training and this improvement is significantly better across training. This allows for another interpretation of the method outside of cross-entropy loss.

## G   Test Accuracy across Training

We plot accuracies over training as a complement to cross-entropy loss in fig. 11 for the sequence modeling experiments. We can use these experiments to de-couple our results from properties of cross-entropy loss that may lead to misleading improvements across training. We find that accuracies improve consistently across training, supporting the loss curve interpretation that guided training improves results.

## H   Basic Distillation Comparison

To show the effectiveness of guidance, we compare it with distillation from [33]. Distillation involves transferring knowledge from a performant teacher network to a less performant student network via maximizing the alignment of the output logits. This encourages the student to have similar predictions as the teacher network. This occurs via the following loss function. Assume $\boldsymbol{Q}$ is the logits extracted from the target (student) network and the $\boldsymbol{P}$ is the logits extracted from the guide (teacher) network.

$$\mathcal{L}_{\text{distill}} = \alpha * T^2 * \text{KL}(\sigma(\boldsymbol{Q}/T)||\sigma(\boldsymbol{P}/T)) + (1 - \alpha) * \mathcal{L}_{\text{CE}}(\boldsymbol{Q}, y) \tag{8}$$

where $y$ is the ground truth labels, $T$ is the temperature to soften the logits, and $\alpha$ is the weighting factor between the distillation loss and cross-entropy loss. In this case, KL refers to the Kullback-Liebler divergence and $\sigma$ corresponds with the softmax function. In practice, we set $\alpha$ to 0.5 and $T$ to 2. We continue to track the full cross-entropy loss across training as well.

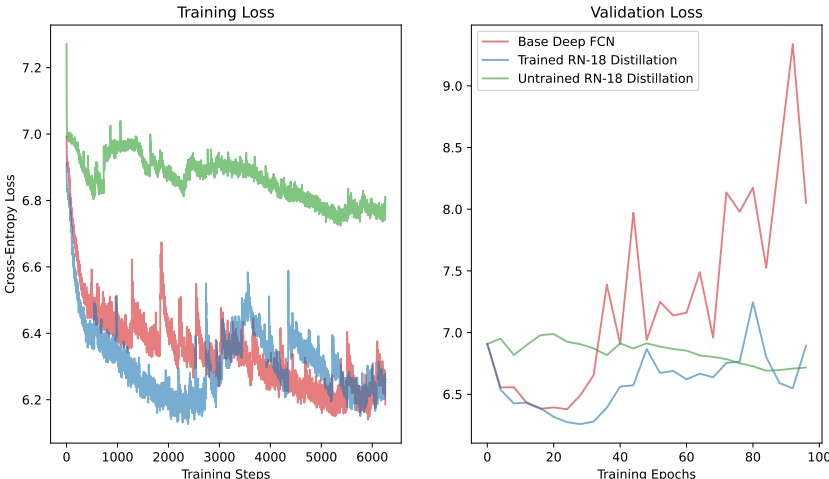

Figure 12: **Distillation does not prevent overfitting**. We compare basic distillation [33] to see if we can prevent overfitting. We use Deep FCN as our student network. We find that distillation with a trained ResNet-18 teacher network leads to a small improvement in performance but still has some patterns of overfitting. Distillation with an untrained ResNet-18 teacher network hurts performance.

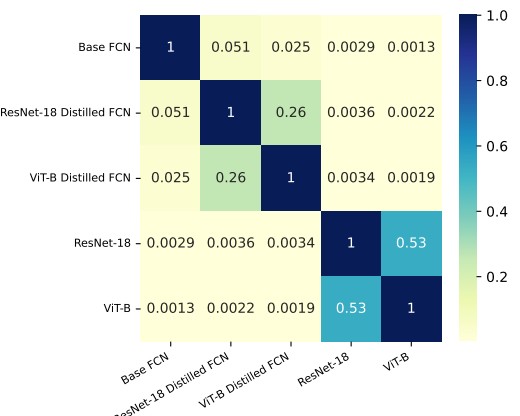

Figure 13: **Distillation lowers error consistency**. In general, we find that distillation results in error consistency patterns that are less consistent than what is reported with guidance.

We show accuracy-based results in fig. 5 and loss curve results in fig. 12. Distillation with a trained network can improve performance but much less than guidance. Distillation with an untrained network reduces performance on average, although not by a significant amount.

## H.1 Error Consistency

We show error consistency performance over distilled networks rather than guided networks in fig. 13.

# I Guided Network Analysis and Interpretation

The results from guidance open many questions in order to explain why untrained guide networks can be better at improving target network performance. We provide an intuitive explanation as well as some geometric analysis of guided networks to see if there is a stronger interpretation.

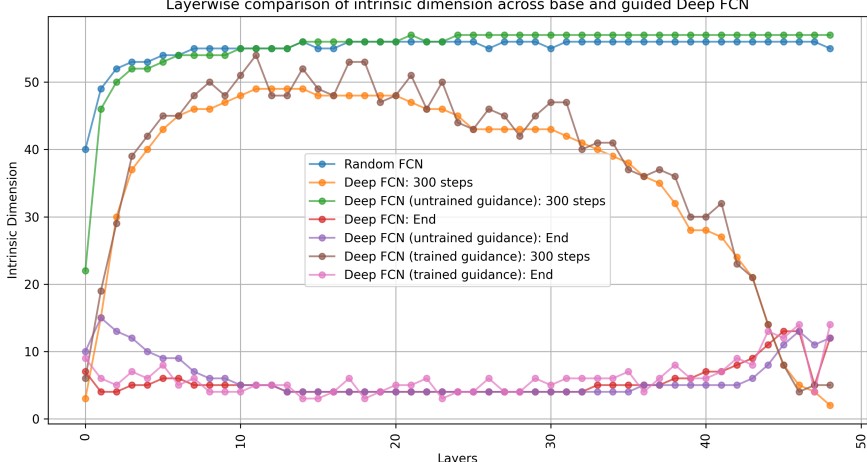

Figure 14: **Guidance preserves intrinsic dimensionality, avoiding over-regularization.** We measure the PCA-based intrinsic dimensionality of representations from each layer of both a guided and unguided Deep FCN at initialization i.e. Random FCN, 300 steps of training, and the end of training. We find that guidance with an untrained guide network better preserves intrinsic dimensionality in comparison to base training or using a trained guide. All networks collapse to the same intrinsic dimensions. This establishes that guidance does change the dynamics of training based on geometric features of the target network.

## I.1 Interpreting Guidance

We believe several prior works support our findings and interpretations in this paper. We cover them here. First, [38] is a recent paper that considers task-aware representational alignment. Their theory provides a generalization bound via kernel alignment. They show that when a "stitcher" maps representations or a source network to a target output, the excess risk of the stitched model is upper-bounded by the CKA alignment between them. This provides a learning-theoretic guarantee that the CKA term in guidance reduces the hypothesis class possibilities seen by an optimizer. Overfitting or underfitting becomes harder.

Shan and Bordelon [70] investigates how a network's neural tangent kernel (NTK) aligns with a target output during training. The paper shows that NTK alignment accelerates convergence and lowers generalization error in deep linear networks. This aligned kernel condition is inserted by hand in guidance. Similarly, [3] uses Rademacher complexity tools to show that alignment of tangent-kernel features onto a small set of task-relevant directions compresses the effective model class. This formalizes the notion of guidance as an automatic regularizer, where task directions are replaced by the guide network settings. Finally, [37] demonstrates that after training, the top singular vectors of a network's hidden activations align with the task target vectors. This empirically supports the layerwise CKA choice in guidance. We believe that CKA bounds the risk or complexity in terms of kernel alignment. The NTK and Rademacher analyses show that alignment shrinks the effective hypothesis space and improves conditioning. This aligns with findings based on singular vectors. We could sharpen this theory by changing the alignment used in guidance e.g. moving from aligning on kernels to aligning on singular vectors or eigenvectors instead. A full PAC-style proof specialized to guidance has not been shown in our paper but we leave this to future work.

## I.2 Geometric Analysis via Intrinsic Dimensionality

We aim to understand how guidance with a randomly initialized guide network differs from a trained guide network. To do so, we compare the representation space of a target network guided by a trained guide network and a randomly initialized guide network using intrinsic dimensionality.

Intrinsic dimensionality (ID) refers to the minimum number of dimensions required to capture the structure or variability in the input data. It represents the true complexity of the data manifold, ignoring noise or redundant dimensions. Previous work has found that neural networks have low ID, capturing data in low-dimensional manifolds [48, 60]. Following [22], for a given threshold $\beta$, the

intrinsic dimension is the $d \in \mathbb{N}$ such that the ratio of explained variance for $d$ dimensions of a $N$ dimensional PCA is above $\beta$:

$$\frac{\sum_{i=1}^{d} var(y_i)}{\sum_{j=1}^{N} var(y_j)} > \beta \tag{9}$$

First, we measure the ID of both unguided and guided Deep FCNs across all layers at different points of training; see fig. 14. Crucially, at 300 steps of training, we notice that guidance with an untrained guide network preserves the initial ID found in the randomly initialized Deep FCN representations. The trained guide network and base training achieve low ID values, which is consistent with findings in prior work [11]. At the end of training, all networks reach the same intrinsic dimension. This finding indicates that the dynamics of training, as shown by ID, change with or without guidance. The Deep FCN without guidance achieves a low ID too early in training, and this is likely similar for the guidance with a trained guide network.

One interpretation of these results is that ID has an effect on overfitting in the Deep FCN. When ID is too low, the Deep FCN overfits. Therefore, we can discover a new regularization scheme for training the Deep FCN based on ID. In this scheme we introduce a new loss function, which designs a differentiable version of PCA-ID based on a specific ID threshold and forces the ID of the representations to be above a particular ID. In particular, given the activations from a specific layer of the target network, $A_i^T$, a variance threshold $\tau$, and a target ID $t$, we first find the SVD of the activations,

$$A_i^T = U \Sigma V^T \tag{10}$$

We extract the eigenvalues using the singular values, $\lambda_j = \Sigma_{jj}^2$ and find the total variance using the eigenvalues.

$$T = \sum_{j=1}^{r} \lambda_j + \epsilon \tag{11}$$

where $r$ is the total number of nonzero singular values and $\epsilon$ is a small constant for numerical stability. Afterwards, we find the explained ratios and cumulative sums of the eigenvalues:

$$p_j = \frac{\lambda_j}{T} \ , \ c_k = \sum_{j=1}^{k} p_j \tag{12}$$

We use the cumulative sum to find the loss for being below the target ID in eq. (13).

$$\ell_{\text{below}} = \sum_{j=1}^{t-1} \sigma(\beta(c_j - \tau)) \tag{13}$$

where $\sigma$ is the sigmoid function and $\beta$ controls the sharpness of the loss. Similarly, eq. (14) gives the loss for being too far above the target ID.

$$\ell_{\text{above}} = \sigma(\beta(\tau - c_t)) \tag{14}$$

Our total loss is given by $\mathcal{L} = \ell_{\text{above}} + \ell_{\text{below}}$. Using this loss to control ID, we tune each layer in our Deep FCN and see a final validation loss given in fig. 15. We find that using our new loss function to control the ID of every linear layer of our Deep FCN leads to improved validation performance and an accuracy of 14.65%. This aligns with guidance, indicating that guidance may be controlling the ID of the target network. Furthermore, we have used guidance to find a new regularization scheme based on intrinsic dimensionality.

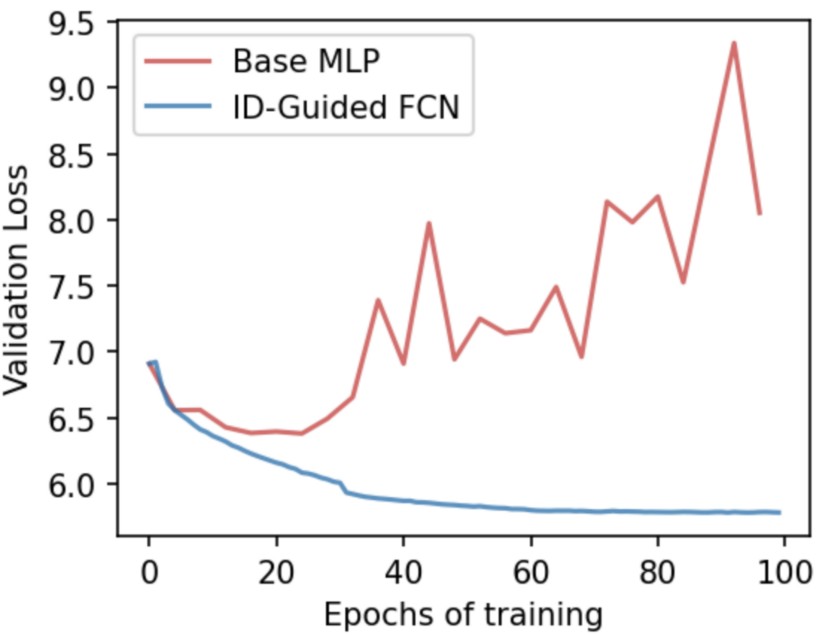

Figure 15: **Controlling intrinsic dimensionality of Deep FCN representations improves image classification performance.** We introduce a novel loss function to regularize the ID of Deep FCN representations in each linear layer of the network during training. We find that this leads to improved validation accuracy, similar to guidance. This shows that guidance correlates with geometric modifications to target networks and can find new regularizations.

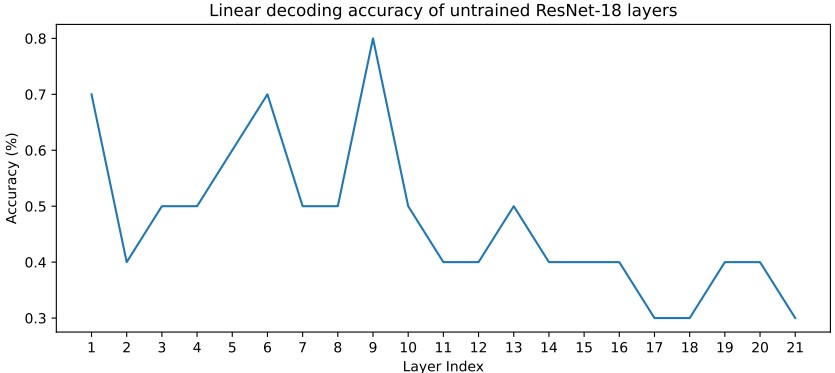

Figure 16: **ImageNet classes are barely decodable from a randomly initialized ResNet-18.** In order to assess the performance of our randomly initialized networks, we design a linear decoder to decode ImageNet classes from all layers of the network. Chance accuracy is 0.1% (1/1000) on the graph above. We find that, while we can decode ImageNet classes with an accuracy above chance, the accuracy is still very low.

### I.3 Linear Decoding

We assess whether object classes are decodable from internal representations of a randomly initialized ResNet-18. A potential explanation for improvements in the target network with an untrained guide is the ability to linearly decode the ImageNet classes with significant accuracy at certain layers of the untrained guide network.

We train a linear decoder with 4000 ImageNet images from the train set and test on 1000 images from the validation set for each layer. We show results in fig. 16. We find that the linear decoder never achieves any accuracy greater than 0.8% for any of the layers. Furthermore, later layers contain little information that is useful to linearly decode ImageNet classes. This means that linearly decodable information isn't present in the guide network and this aspect isn't driving improvements in target networks. We note that this matches findings in Amid et al. [2], which reports that linear decodability from a ResNet-18 achieves 3.4% top-1 accuracy. The increase in performance is likely due to using a much larger dataset.

## J  Guidance with RSA and Ridge Regression

### J.1  Representational Similarity Analysis

We use the RSA formulation as described in Kriegeskorte et al. [46]. Specifically, RSA constructs representational dissimilarity matrices (RDMs) for two sets of representations and compares them using an outer similarity function.

Given two sets of representations, $\boldsymbol{R} \in \mathbb{R}^{b \times d_1}$ and $\boldsymbol{R}' \in \mathbb{R}^{b \times d_2}$, we first calculate RDMs for each set of representations using a distance function $d$. Formally, we define $\boldsymbol{D} \in \mathbb{R}^{b \times b}$ as

$$\boldsymbol{D}_{i,j} := s(\boldsymbol{R}_i, \boldsymbol{R}_j) \tag{15}$$

Each row $\boldsymbol{D}_i$ corresponds to the distance between the representations of input $i$ and the representations of all inputs including itself. This is done per-batch, meaning that RSA is sensitive to batch size.

Given two RDMs $\boldsymbol{D}$ and $\boldsymbol{D}'$ constructed from sets of representations $\boldsymbol{R}$ and $\boldsymbol{R}'$ respectively, we vectorize the RDM matrices using a function $v$ (since the RDMs are symmetric, we only need to compare the lower triangles), and compute the similarity between the two vectorized RDMs using a similarity function $s$.

$$\mathcal{M}(\boldsymbol{R}, \boldsymbol{R}') = s(v(\boldsymbol{D}), v(\boldsymbol{D}')) \tag{16}$$

As with CKA, we use the complement of the similarity to construct $\bar{\mathcal{M}}$. In practice, we define $d$ to be the cosine distance between every pair of inputs and $s$ to be the pearson correlation between the RDMs as done in previous work [14].

We apply guidance with RSA to Deep FCN as our target network and ResNet-18 as our guide network. Similar to our CKA results, we train for 100 epochs with a batch size of 256, as RSA is sensitive to the number of samples when comparing sets of representations.

### J.2  Ridge Regression

We used similar ridge regression formulation as [14, 71] without cross-validation.

Given two sets of representations, $\boldsymbol{R} \in \mathbb{R}^{b \times d_1}$ and $\boldsymbol{R}' \in \mathbb{R}^{b \times d_2}$, we first apply a sparse random projection on the representations. Since the dimensionality of the representations is prohibitively large, the projection makes the ridge regression feasible to compute. We refer to the resulting representations as $\boldsymbol{P}$ and $\boldsymbol{P}'$ which correspond to the projected representations $\boldsymbol{R}$ and $\boldsymbol{R}'$ respectively. $\boldsymbol{P}$ and $\boldsymbol{P}$ have dimension $d$, where $d$ is fixed using the Johnson-Lindenstrauss lemma [40].

Afterwards, we mean-center the representations and apply ridge regressions using the original least-squared solution as follows. Our goal is to predict the representations $\boldsymbol{P}'$ using regressors over $\boldsymbol{P}$. We first split our representations into a training set i.e. $\boldsymbol{P}_{\text{train}}, \boldsymbol{P}'_{\text{train}}$ and testing set $\boldsymbol{P}_{\text{test}}, \boldsymbol{P}'_{\text{test}}$ where

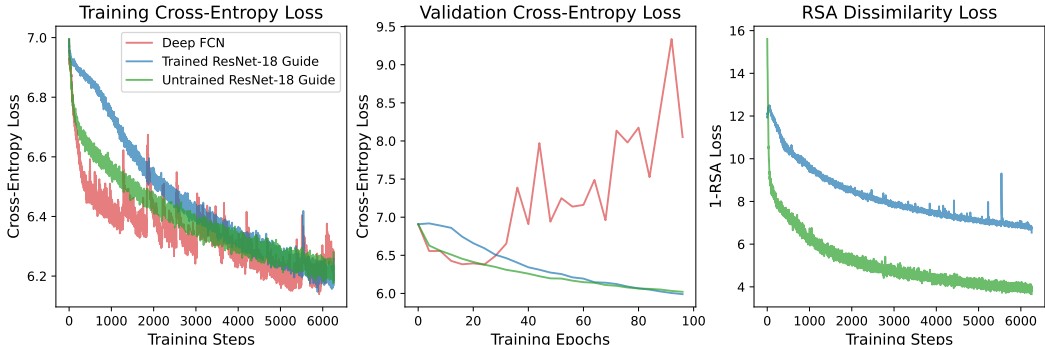

Figure 17: **Guidance with RSA as the representational similarity metric maintains similar performance to CKA**. We include a further experiment where we change the metric for representational alignment from CKA to RSA during guidance training. We apply this to the Deep FCN with ResNet-18 as a guide network. We see that, like CKA, RSA alignment also allows for transferring the prior from ResNet-18. However, unlike CKA, the untrained guide network only does marginally better than the trained network, potentially indicating the RSA is better at transferring trained features.

the training set contain half the representations and the testing set contains the other half. We first a set of regressors $\hat{\beta}$ as follows:

$$\hat{\beta} = ((\boldsymbol{P}_{\text{train}})^T \boldsymbol{P}_{\text{train}} + \lambda \boldsymbol{I}_d)^{-1} (\boldsymbol{P}_{\text{train}})^T \boldsymbol{P}'_{\text{train}} \tag{17}$$

where $\lambda$ is the ridge penalty, which is a hyperparameter. The coefficients $\hat{\beta}$ are then used to predict the held out data where:

$$\hat{\boldsymbol{P}'_{\text{test}}} = \boldsymbol{P}_{\text{test}} \beta \tag{18}$$

We measure the cosine similarity between the predicted representations $\hat{\boldsymbol{P}'_{\text{test}}}$ and actual representations $\boldsymbol{P}_{\text{test}}$.

$$\mathcal{M}(\boldsymbol{R}, \boldsymbol{R}') = \text{cosine}(\hat{\boldsymbol{P}'_{\text{test}}}, \boldsymbol{P}'_{\text{test}}) \tag{19}$$

We apply guidance with RSA to Deep FCN as our target network and ResNet-18 as our guide network. Similar to our CKA results, we train for 100 epochs with a batch size of 256, as ridge regression is sensitive to the number of samples when comparing sets of representations. We manually tune the $\lambda$ hyperparameter, finding that $\lambda = 10.0$ is optimal for the trained guide network and $\lambda = 100.0$ is optimal for the untrained guide network.

### J.3 Results

**RSA**: We see results over the training, validation, and representational dissimilarity loss in fig. 17. The Deep FCN guided by a trained ResNet-18 achieves an accuracy of *11.02%* and the Deep FCN guided by a randomly initialized ResNet-18 achieves an accuracy of *11.74%*.

We can first observe that guided training improves over base training as noted in fig. 2 and table 3. This demonstrates the generality of our approach to other metrics. As long as a representational similarity metric is differentiable, we can optimize the metric for alignment between two networks as a method to transfer the prior of one network to another.

We can also observe some minute differences between the results with CKA. Most notably, the trained guide network has similar performance to the untrained guide network. This is likely because less information about trained features are present in the RSA metric. RSA measures relative distance between input instances and imposes a constraint of placing these into relative distances. It could be possible that fewer degrees of freedom are useful for aligning target network with trained guides.

**Ridge**: We see results over the training, validation and representational dissimilarity loss in fig. 18. The Deep FCN guided by a trained ResNet-18 achieves an accuracy of *9.46%* and the Deep FCN

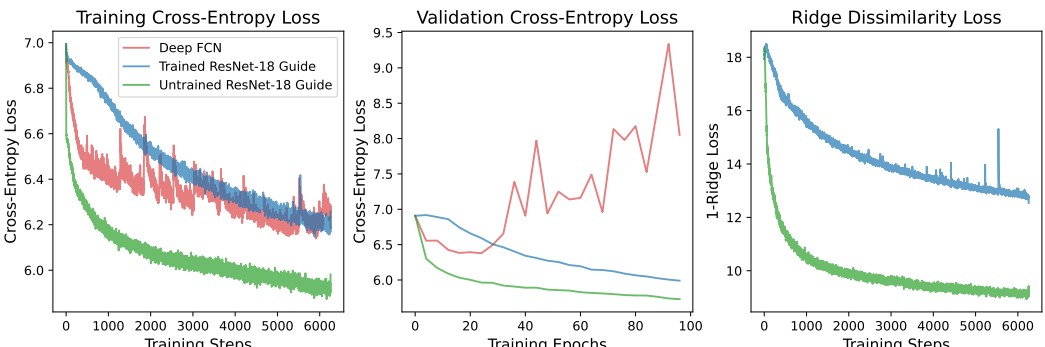

Figure 18: **Guidance with ridge regression as the representational similarity metric improves performance over CKA**. We change the metric for representational alignment from CKA to ridge regression during guidance training. We apply this to the Deep FCN with ResNet-18 as a guide network. We see that, like CKA, ridge regression alignment also allows for transferring the prior from ResNet-18. We find that this improves over CKA significantly

| Experiment | CIFAR-10 Test Accuracy (↑) |
|---|---|
| Deep FCN | 60.58 |
| All layer Guidance | 70.15 |
| Last layer Guidance | 67.18 |
| Last two layers Guidance | 68.03 |
| Last five layers Guidance | 67.50 |
| Last ten layers | 69.31 |
| First ten layers | 79.22 |
| First five layers | **79.58** |
| First two layers Guidance | 73.34 |
| First layer Guidance | 65.11 |
| Multiple Guide Layers | 68.14 |

Table 5: **Guiding earlier layers of deep networks leads to better results**. We apply an ablation experiment to identify which layers lead to stronger improvement when guided. We use a Deep FCN, guided by a randomly initialized ResNet-18 on CIFAR-10. We find that guiding earlier layers leads to strong improvement, even over guiding all layers. Guiding any layer leads to an improvement of performance.

guided by a randomly initialized ResNet-18 achieves an accuracy of *15.69%*. Similar to RSA and CKA, we can see the guided training improves over base training.

Similar to CKA, we observe that randomly initialized guide networks outperform trained guide networks. Furthermore, performance with ridge regression is better than with CKA. This finding is intuitive. Ridge regression generally has more degrees of freedom than other similarity metrics because of fewer invariances imposed on the metric. This means that the solution search space is larger, leading to better results. We believe this provides a promising path forward for making target networks have better performance.

Furthermore, ridge regression has other desirable properties such as potential explainability via probing predicted representations to measure similarity. We can use probing analyses on predicted representations to see what information the target has inherited from the guide network. This opens up many avenues for studying guidance in the future.

| Experiment | Copy-Paste Accuracy ($\uparrow$) |
|---|---|
| RNN | 14.35 |
| All layer Guidance | **42.56** |
| Last layer Guidance | 38.19 |
| Last three layers Guidance | 42.33 |
| Last two layers Guidance | 41.55 |
| First layer Guidance | 27.59 |
| First two layers Guidance | 28.15 |
| First three layers Guidance | 33.93 |
| One guide layer | 36.11 |

Table 6: **Guidance of later layers improves RNN performance**. We apply an ablation experiment over RNNs trained for copy-paste to see whether guiding certain layers lead to improved performance. We find that guiding later layers leads to stronger performance overall. Furthermore, RNN layers are guided by several guide network layers in the transformer such as the linear layer and layer-normalization in the transformer decoder. Including both of these leads to better results.

## K Ablation Experiments

We analyze the current design of our layer mapping for guidance by experimenting with the number of layers used in guidance and whether more complex mappings exist like mapping several layers of the guide network to a single target network.

We run layer-wise ablation experiments the Deep FCN guided by an untrained ResNet-18 over CIFAR-10. Similarly, we experiment with RNNs guided by untrained transformers over the copy-paste task.

In table 5, we first show the effect of guiding over a subset of layers in the Deep FCN, evaluated over CIFAR-10. We find that earlier layers are much more impactful for a deep network. Intuitively, this could be due to guidance providing aiding with the credit assignment problem in deep networks: gradients don't propagate properly to earlier layers. However, table 6 shows that later layers in the RNN are more useful to apply guidance to when improving copy-paste performance. In general, we find that guiding any layer leads to improvements in results generally, showing the general applicability.

Furthermore, in table 5 and table 6, we consider new methods to map guide network layers to target network layers. When guiding a Deep FCN with ResNet-18, we only apply a 1-1 layer mapping for supervision i.e. each Deep FCN layer is guided by only one ResNet-18 layer. From table 5, introducing multiple sources of supervision from guide network layers by allowing a one-to-many mapping decreases performance. However, with the RNN, we guide with representations from the linear layer and layer normalization in the transformer decoder. One could consider that linear layers are redundant with layer normalization for guidance, so we remove its representations as a potential supervisory target. We find that this hurts performance, showing that RNNs benefit from multiple levels of supervision from its guide.

Understanding the dynamics of guidance supervision is interesting and could allow for understanding training dynamics of neural networks or allow us to form cross-architectural relationships.

