# OpenReview forum: "Training the Untrainable: Introducing Inductive Bias via Representational Alignment"
_NeurIPS.cc/2025/Conference — NeurIPS 2025 poster_

### Official Review · Reviewer_xZWb · 2025-06-23

**Clarity:** 2
**Significance:** 3
**Originality:** 3
**Rating:** 4
**Confidence:** 4

**Summary:**

The paper investigates whether models traditionally deemed unsuitable for certain tasks can perform well when guided by inductive biases from more suitable architectures. The proposed method employs a neural distance function and layerwise representational similarity to guide the "unsuitable" model. The authors conduct experiments across several architectural pairs, including FCN vs. CNN, RNN vs. Transformer, and No-skip-CNN vs. ResNet. Interestingly, they observe that even randomly initialized (untrained) models can serve as effective guides, leading to performance improvements in the target model.

**Questions:**

1. The authors claim that "Guidance can use a smaller untrained guide instead of a larger trained teacher." Would using a larger trained guide network further enhance the performance of the target network compared to using a smaller one?
2. Have the authors considered aligning feature statistics (e.g., mean and variance) [1] during representation matching?

Ref.

[1] Mirza et al., "ActMAD: Activation Matching to Align Distributions for Test-Time-Training", CVPR 2023

**Ethical Concerns:**

["NO or VERY MINOR ethics concerns only"]

**Final Justification:**

I appreciate the authors’ efforts in the rebuttal, and my overall concerns have been addressed. However, since some revisions are still needed, my score remains unchanged.

**Limitations:**

Yes

**Paper Formatting Concerns:**

I do not notice any major formatting issues.

**Quality:**

3

**Strengths And Weaknesses:**

Pros:
1. The paper introduces the concept of an inductive bias gap between modern neural network architectures and proposes bridging this gap through representational alignment. The method is simple, yet the findings are surprising.
2. The experiments are extensive and systematic, validating the effectiveness of the proposed approach.
3. The method presents a useful framework for studying architectural priors, with potential applications in neural network design and initialization.


Cons:
1. Some claims in the paper should be toned down. For example, the statement about "providing theoretical insights"  (line 62) of the introduction seems overstated given the mainly empirical nature of the paper.
2. Some of the tables should be improved. In Table 1, consider reorganizing the layout to help readers easily identify which experiments involve "untrainable" architectures versus "untrainable" tasks. In Table 2, highlight the target networks using different background colors for better readability. In Table 5, include the Deep FCN results from Table 2 to improve readability and allow for easier comparison.
3. Given that distillation is a natural and well-established baseline, why was it not thoroughly compared and included in Table 2?
4. The explanation regarding why untrained guide networks sometimes yield better performance in Table 2 is not fully convincing. In Appendix E, the Wide FCN paired with untrained ResNet shows lower similarity loss, yet paired with the trained ResNet achieves higher accuracy.

---

> ### Author Rebuttal · Authors · 2025-07-30
>
> We thank the reviewer for their thoughtful review. We are glad the reviewer found the ideas in the paper novel, the method simple, and the results surprising. We hope to address some concerns below.
>
> > W1: Some claims in the paper should be toned down. For example, the statement about "providing theoretical insights" (line 62) of the introduction seems overstated given the mainly empirical nature of the paper.
>
> This is fair. We will aim to tone some of the claims in the paper, specifically noting guidance’s ability to differentiate between structural and knowledge-based priors.
>
> > W2: Some of the tables should be improved. In Table 1, consider reorganizing the layout to help readers easily identify which experiments involve "untrainable" architectures versus "untrainable" tasks. In Table 2, highlight the target networks using different background colors for better readability. In Table 5, include the Deep FCN results from Table 2 to improve readability and allow for easier comparison.
>
> Thank you. These are great suggestions. We will absolutely fix the layout of Table 1 and 2 to better indicate the understanding of untrainable architectures and tasks as well as have a better coloring scheme to indicate performance. Such changes will also go to Table 3. We really appreciate the help on readability.
>
> > W3: Given that distillation is a natural and well-established baseline, why was it not thoroughly compared and included in Table 2?
>
> This is fair. We will include the numbers from the appendix into the main paper upon acceptance. This will be an additional set of rows, copied here for simplicity. We will modify the table to match your suggestions on formatting as well.
>
> | Experiment                                | Copy-Paste Accuracy | Parity Accuracy | Language Modeling Perplexity |
> |-------------------------------------------|---------------------|-----------------|------------------------------|
> | RNN Teacher/Transformer Student           | N/A                 | 75.11           | N/A                          |
> | Untrained RNN Teacher/Transformer Student | N/A                 | 68.98           | N/A                          |
> | Transformer Teacher/RNN Student           | 19.61               | N/A             | 52.00                        |
> | Untrained Transformer Teacher/RNN Teacher | 14.33               | N/A             | 73.16                        |
>
> > W4: The explanation regarding why untrained guide networks sometimes yield better performance in Table 2 is not fully convincing. In Appendix E, the Wide FCN paired with untrained ResNet shows lower similarity loss, yet paired with the trained ResNet achieves higher accuracy.
>
> This is a fair point, the reviewer is right that our initial explanation in the paper was completely thorough. We will add some discussion here and update our paper to address these points. We believe the inconsistency with the Wide FCN is likely due to properties of wide neural networks as well as the systemic failure of the Wide FCN. For the Wide FCN, an untrained ResNet-18 is easier to mimic. This is likely due to the simplicity of the initialization of an untrained ResNet-18 as well as properties of very wide linear layers as universal function approximators. The lower CKA distance with the untrained guide network indicates strong architectural transfer. But the lower accuracy indicates that the feature transfer is not task aligned i.e. the learned filters are more relevant for the Wide FCN. Intuitively, the Wide FCN already generalizes reasonably, and wide linear layers can approximate convolutional layers [1]. So the remaining gap is feature quality, not architectural gaps. Therefore, a trained guide adds class selective filters that a random guide cannot provide.
>
> One other explanation for our performance with untrained guides is that target networks that overfit can be aided with a randomly initialized guide network. Targets that lack discriminative features like the Wide FCN may be aided by a trained guide that adds semantic bias. We will aim to make this more clear in the paper.
>
>
> > Q1: Would using a larger trained guide network further enhance the performance of the target network compared to using a smaller one?
>
> This is a great question. We actually find that a larger, trained guide network does not perform as well as a smaller guide most of the time. Our intuitive explanation for this is that the target network does not have the capacity to learn useful information from a larger guide network. We find that having a sufficient number of parameters is important for transferring knowledge and priors from the guide network. For example, if we attempt to guide a 4-layer RNN with a trained 6-layer transformer for language modeling, our final perplexity is actually 52.55 which is worse than the score reported. The RNN does not have sufficient capacity to learn the same function as the guide network. We hope this shows that what we can learn from guidance goes beyond distillation. We have now found out a relationship between the parameter count of an RNN and how much is sufficient to learn a function that a specific transformer can learn.
>
> > Q2: Have the authors considered aligning feature statistics (e.g., mean and variance) [1] during representation matching?
>
> We did not consider this, thank you for the suggestion. This could validate whether our intuitions about CKA are correct. CKA is a measure that depends on second-order statistics, specifically distance matrices across samples. Architectural choices imprint distinct features on those statistics. For example, consider local receptive fields in a convolutional layer. Units that cover neighbouring pixels receive correlated inputs, and this is reflected in our activations. Such correlations are reflected in our distance matrices, and these can be transferred to distance matrices associated with FCN layers that lack local correlations. It’s unclear whether the same is captured by simple mean and variance statistics. One concern is that the mean does not capture correlated inputs well enough, and this could lead to potential failure. However, it would be exciting if even the mean and variance capture enough information to influence how an FCN or RNN layer is activated, and this could have strong implications for understanding alignment and inductive biases of architectures in the future. Due to the time constraints of this rebuttal, we could not include this experiment but will go deeper into this direction in the future.
>
> [1] Malach and Shalev-Shwartz Computational Separation Between Convolutional and Fully-Connected Networks, ICLR 2021.

---

> ### Comment · Reviewer_xZWb · 2025-08-08
> **Official Comment by Reviewer xZWb**
>
> My overall concerns have been addressed. I remain positive.

---

### Official Review · Reviewer_Tqwd · 2025-06-25

**Clarity:** 1
**Significance:** 2
**Originality:** 1
**Rating:** 3
**Confidence:** 4

**Summary:**

This paper proposes to apply a guidance network to train a target network, the intermediate representations between the two networks are aligned. Thus, the target network could be less overfitted.

**Questions:**

Please see the weaknesses above.

**Ethical Concerns:**

["NO or VERY MINOR ethics concerns only"]

**Final Justification:**

After the discussion, I still believe the proposed method is utilizing knowledge distillation with hand-crafted features, though new names are assigned to those un-trained networks as architectual biases. I changed my rating, but this is still below acceptance to me at this moment.

**Limitations:**

I would suggest to better clarify the purpose of ths draft, if it tries to better regularize the model, then introduce the method and compare against other regularization methods comprehensively, the pros and cons, the performance and cost. If this draft intends to propose a better disllation process, then rethink the target task accordingly. I would encourage a major modification before resubmitting.

**Quality:**

1

**Strengths And Weaknesses:**

The clarity of this draft needs improvement, some points are not clear enough to me.

1. Line 2 “We call a network untrainable when they overfit, underfit..” This definition seems very vague to me. A model could be under- or over-fitted due to numerous reasons, but they are all trainable. And model regularization is a long-standing topic, maybe this study falls into this bucket.

2.Line 10 “If the guide is trained, … If the guide is untrained,...” There exist different model architectures, some well-designed models can learn the inductive bias better. If the model is “untrained”, personally I don’t believe the model contains “the” inductive bias we want for down-stream models or tasks. We can discuss a little why the untrained may also slightly work later in this thread.

3. Line 16 “Our method provides a mathematical tool…”, I can only find some empirical findings in the experiment based on the widely used loss and accuracy, is this the new tool provided in this draft? Please correct me if I am wrong.

4. I can see the terms “mitigate”, “stop” overfitting occurred many times, personally, I highly doubted if the distillation-like solution is necessary for this purpose. There are different ways to prevent overfitting on network, simple dropout or l2 regularization. However, I didn’t see any mathematical discussion nor empirical comparison on different regularization. Which may deliver “cheaper” solutions for overfitting.

5. Is it novel that the finding of using a guidance network can prevent overfitting? This is not very surprising. Given the guidance network is not overfitted, then aligning the internal representation between the two models will definitely “compromise” the guidance model. And this is core assumption behind distillation.

6. Line 121, “We distinguish guidance from distillation…”, distillation can be applied to either the output or the intermediate features, I didn’t see any difference between this draft and other distillation studies based on Eq1, though the purpose could be slightly different. I won’t be surprised if a large trained teacher can do a better job for this purpose, given the better teacher learns a better representation which can better generalize on the validation data, then “compromising” on it can also deliver a better validation loss(less overfitted).

7. Following the question in Q2, is the use of “untrained” model for a guidance novel? Some models maybe initialized as a Gaussian filter, which could be roughly used as a Gabor filter back to the era of hand-crafted features. Again, comprising on this un-fitted initialization can lead to a less over-fitted curve, but this procedure is meaningless. Aligning with unlearned filters will also lead to an under-fitted target network, which makes the use of untrained models unnecessary, the target model can not properly fit on any thing, tuning the loss weight is trivial in this case.

To sum up, applying distillation (I still believe the trick used in this draft is the same as the common distillation) singly for over-fitting seems not necessary, I belive other solutions can do the same job more efficiently. Distillation can be used to train a better target model for sure, and that superiority of the target model also includes less-overfitting.

---

> ### Author Rebuttal · Authors · 2025-07-30
>
> We thank the reviewer for their thoughtful review. We hope to address concerns on the clarity of our draft and the aim of our paper.
>
> > W1: A model could be under- or over-fitted due to numerous reasons... And model regularization is a long-standing topic...
>
> We apologize if our definition seemed vague. We specifically called architectures trainable when their performance was near-chance. For our networks, this near-chance performance is driven by overfitting, underfitting, or some issue with incorporating state or memory. This is also based on discussions from prior work, see [1, 2, 3, 4]  which connect overfitting and underfitting to trainability. [1] states that “Deeper neural networks are more difficult to train”. In [1], SGD cannot find a low-error solution even though one exists by construction. That is the very definition of an untrainable architecture. [2] treat overfitting as a core obstacle that must be fixed for a model to be usable. [3], [4] and many other papers talk explicitly about improving trainability by fixing either optimization instability (underfitting) or excessive capacity (overfitting). Our goal was to take such architectures that have near-chance performance and show that we could achieve performance significantly above chance, even obtaining an architecture that was potentially usable.
>
> Our paper aims to be more general than focusing entirely on model regularization. Our current focus is on failures driven by architecture that cannot be solved by current optimization or regularization techniques. Guidance with untrained networks could discover new regularization strategies that could not be uncovered before. However, our overall goal is to discover when neural networks failures are due to difficulties with knowledge priors, structural priors, or initialization.
>
> > W2: … personally I don’t believe the model contains “the” inductive bias we want for down-stream models or tasks.
>
> We respectfully disagree. Prior work and our paper show that untrained networks contain a great deal of information that is useful for a downstream task. For example, [5] states that the structure of a CNN generator network is sufficient to capture a great deal of low-level image statistics prior to any learning and yields competitive denoising and inpainting performance with untrained networks. Convolutions impose local receptive fields, weight sharing, and translation equivariance, all of which shape our distance matrices as part of CKA. Residual connections preserve gradient flow. Those structural properties show up immediately in the gram matrices that CKA compares, which is why an untrained ResNet guides an FCN so effectively. Self-attention also imposes inductive biases useful for modular arithmetic [6]. Our paper provides further empirical proof of this.
>
> Trained models certainly embed both architectural and learned biases that may capture a task’s structure more cleanly. But that does not negate the fact that architecture alone already encodes a nontrivial prior in most cases. One major point of guidance is to isolate architectural bias from learned bias, a distinction methods like distillation cannot make. Results with untrained guide networks would not work if an untrained network truly lacked inductive bias that was relevant for downstream models or tasks.
>
> > W3: Line 16 “Our method provides a mathematical tool…”, is this the new tool provided in this draft?
>
> We will revise the language to characterize our claim better. We believe guidance really is a mathematical tool rather than just an accuracy trick, unlike distillation. CKA is a differentiable kernel-alignment metric that is scale invariant and agnostic to labels. Minimizing CKA lets you clamp any subset of a target network’s activations onto those of a frozen guide network and sweep that clamp across architectures and priors and observe what fails or succeeds. Guidance is a knob you can turn to inject architectural priors at will, something one cannot do with cross-entropy or weight-decay alone. We use this in concrete ways such as in Appendix K where we investigate the distinction between early vs late priors to show that early ResNet layers matter most for FCNs or later transformer layers matter most for RNNs. In Table 2 and 3 as well as Appendix E, we show a distinction between architectural and learned priors to isolate pure architectural bias. We gave a formal method for discovering initializations of networks with this alignment, showing that the guidance “clamp” created a better weight basin. Initialization is extremely relevant given the random nature of discovering initialization methods overall. And, as shown in responses to other reviewers (CY31), kernel-alignment theory shows that forcing two models to share the same hidden-feature kernel tightens generalization bounds and improves conditioning. Guidance, to us, is a controllable, differentiable kernel-alignment feature clamp that lets researchers dissect architectural priors and is a regularizer to investigate what priors are helpful.
>
> > W4: There are different ways to prevent overfitting on network, simple dropout or l2 regularization.
>
> Thank you. We include baselines comparing our Deep FCN optimized with L2 regularization and AdamW to guidance. We report the results below:
>
> |Setting |Accuracy |
> |---|---|
> | DFCN + L2  	| 6.4 |
> | DFCN + AdamW   | 7.2 |
> | Guided DFCN	| 13.1 |
>
> We cannot report any loss curves here. We can confirm that L2 regularization prevents some overfitting but not as well as guidance.
>
> > W5: Is it novel that the finding of using a guidance network can prevent overfitting? This is not very surprising.
>
> We hope to make this more evident. First, the focus of our paper is not only on overfitting. We consider networks that struggle to incorporate memory like RNNs or underfit like Deep ConvNets or Deep FCNs. We believe having a general purpose method in all of these cases is novel. We would also like to point out that across all of settings, we see major improvements. We take non-functional architectures like FCNs or RNNs and turn them into functional architectures by transferring structural priors. This is not reproduced by logit-based distillation either.
>
> With regards to regularization, we also see guidance and our paper as novel. Most regularizers prevent overfitting by shrinking weights, adding noise, or transferring task targets. Here, architectural patterns alone restore generalization. Deep FCN, deep plain CNNs, RNNs, are all known to overfit or underfit despite tuning, identity/orthogonal inits, dropout, weight decay, label-smoothing, and distillation. In our paper, guidance is the first single technique shown to rescue all three classes of pathological failures without changing architecture or adding train time augmentations.
>
> > W6: I won’t be surprised if a large trained teacher can do a better job for this purpose...
>
> In our paper, many of the guide networks are smaller in terms of parameter count and layer count. We find that larger guide networks do not necessarily do a better job. The target network does not have the capacity to learn useful information from a larger guide network like knowledge and structural priors. If we attempt to guide a 4-layer RNN with a trained 6-layer transformer for language modeling, our final perplexity is actually 52.6 which is worse than the score reported. The RNN does not have the capacity to match the guide network. This shows that what we can learn from guidance goes beyond distillation. We have now found out a relationship between the parameter count of an RNN and how much is sufficient to learn what a specific transformer can learn.
>
> > W7: Following the question in Q2, is the use of “untrained” model for a guidance novel...?
>
> This is not what we empirically find in our paper. If alignment with unlearned filters merely “compromised” the target, we would expect little change in training loss and lower test accuracy. This paper observes the opposite: training loss improves while the generalization gap closes. For instance, 4- and 6-layer RNNs improve with an untrained transformer, achieving LM perplexities that are competitive with LSTMs. This shows proper fitting. Underfitting is not theoretically guaranteed either. [7] and [8] prove that matching the feature kernels reduces complexity while preserving optimal risk. In our setting, this provides evidence that we are shrinking the function class just enough to avoid memorization while ensuring a solution is reachable.
>
> We also argue that guidance is distinct from Gaussian/Gabor-like filters. A single Gaussian filter is a hand-crafted prior that is applied to the first layer. A randomly initialized guide such as ResNet contains dozens of layers of convolutions, residuals and batchnorm stats. This stack must impose priors like locality and receptive fields, residual spectra, etc. CKA is sensitive to this structure, not just first layer edge filters.
>
> We believe the novelty of our results agrees with findings in [5] where authors show that random filters carry signal. We are the first, to the best of our knowledge, to use these filters as an external tool to fix a different architecture. Our comparison against distillation showed that distillation needed a trained teacher.
>
> [1] He et. al. “Deep Residual Learning for Image Recognition”, CVPR 2016.
> [2] Srivastava et al. “Dropout: A Simple Way to Prevent Neural Networks from Overfitting”. JMLR 2014.
> [3] Ioffe & Szegedy. “Batch Normalization”, ICML 2015
> [4] Wang and Fu. “Trainability Preserved Neural Pruning.” ICLR 2023.
> [5] Ulyanov et. al. “Deep Image Prior”. CVPR 2018.
> [6] Zhong and Andreas. “Algorithmic Capabilities of Random Transformers”. NeurIPS 2024.
> [7] Canatar et. al. “Spectral Bias and Task-Model Alignment Explain Generalization in Kernel Regression and Infinitely Wide Neural Networks.” Nature Communication, 2021.
> [8] Bietti and Mairal. “On the Inductive Bias of Neural Tangent Kernels”, NeurIPS 2019.

---

> > ### Comment · Reviewer_Tqwd · 2025-08-08
> > **Further discussion**
> >
> > Thanks the authors for the further discussion.
> > Q1: "For our networks, this near-chance performance is driven by overfitting, underfitting, or some issue with incorporating state or memory", If the trainability in this study is still associated with over-ftting or underfitting, I think the two different concepts are mis-connected together. Thanks for the references, but I cannot find proof in those studies about "Deeper neural networks are more difficult to train" is due to overfitting, acutually, I cannot search the term "trainable" in both ResNet and BN papers. For the other two papers, the concpet of "trainable" parameters matches my understanding, if the parameters are optimized or frozen. Again, I can find many papers about overfitting on networks, but the definition seems orthongonal to the concept of "trainability", e.g., [Understanding deep learning requires rethinking generalization].
> > Q2: "Inductive bias" from [5], thanks for pointing out this study, but it happens to prove my point in the weakness section. In the abstract of [5], "a randomly-initialized neural network can be used as a handcrafted prior", this is why I was suspecting the intialized network used in this paper behaves like a gabor-like feature extractor. My personal understanding on the model inductive bias, is the LEARNED features or patterns from a SPECIFICALLY DESIGNED network (in [5], they were tring to explore the LEARNED bias from a GENERATOR). Otherwise, if the main contribution in this study attempts to show an initialized network can also deliver inductive bias, I cannot say this is completely wrong, but this is quivalent to combining hand-crafted features with network together for a specific task. The hand-crafted feature is believed to be a lossy version of the free-learned feature, and the combination seems not novel. Following study [5], an initialized network can still devlier a little better result due to the hand-crafted feature, which is defined as trainability in this study.
> > Q3: Thanks for clairifying the tool, but the explanation seems recaping again the proposed method, the idea is mainly based on knowledge distillation, then all of the claimed contributions are inherited from KD. "we show a distinction between architectural and learned priors to isolate pure architectural bias. " I believe the "architectural" bias refers to the untrained network in the tables, then it seems giving a new name "architectural bias" to the "hand-crafted" feature presented in [5].
> > Q5: "First, the focus of our paper is not only on overfitting." Sorry that I might mis-understood the focus, because the paper mainly associates "trainability" to "overfitting"(though this is questionable as discussed above), so I thought the proposed distillation-based method tried to showcase the efficacy of regularization. "Here, architectural patterns alone restore generalization." This falls back to the emphasis of the "hand-crafted feature"(architectual bias), the "side-effect" of the hand-crafted feature could be under-fitted.
> > Q6: Following this question, then this study could be something ground-breaking to me, but the experiment and proof may need major change to turn this draft into a new study. Using smaller guidance(teacher) model to help a relatively larger (student) model can do a better job. I admit this would be a very interesting point to me, (very) loosely speaking, this is equivalent to learning from a less informative sub-space (model) to obtain something richer, but currently the draft didnot convincingly show this. The performance of distilling a larger model is worse than a smaller model (assuming the larger model performs better than the smaller model) could be due to engineering design, but I could be wrong.
> > Q7: Novelty, "We also argue that guidance is distinct from Gaussian/Gabor-like filters.", " A single Gaussian filter is a hand-crafted prior that is applied to the first layer.", "We believe the novelty of our results agrees with findings in [5] where authors show that random filters carry signal. We are the first,", Sorry, I didn't do a through comparison of what the initialized network looks like, may be different from a exact Gabor, but the point is that is is hand-crafted feature, as you also agreed in study [5]. Then this paper is the first to utlize the external hand-crafted features for training networks for down-stream tasks? I believe there was a time that this was a hot topic to explore. (just randomly searched one for reference[Local Learning with Deep and Handcrafted Features for Facial Expression Recognition])

---

> > > ### Comment · Reviewer_Tqwd · 2025-08-08
> > > **sorry that I didn't split the paragraph properly, please follow the marks, Q1-5.**
> > >
> > > N/A

---

> > > > ### Author Response · Authors · 2025-08-09
> > > >
> > > > We thank the reviewer for their feedback. We hope to address more concerns here.
> > > >
> > > > > Q1: If the trainability in this study is still associated with over-ftting or underfitting, I think the two different concepts are mis-connected together.
> > > >
> > > > There seems to be a misunderstanding about how trainability should be defined or considered in our paper and what our references aimed to say.
> > > >
> > > > Our definition of trainability is highly practical and is used in relation to whether a network can be optimized for good training and held-out performance. The references we showed all make design choices to make neural networks easier to optimize. [1], a paper we referenced, says “Trainability describes the easiness of optimization of a neural network” in Section 2. We believe this connects to all the papers that describe overfitting and underfitting although not explicitly stated. The first sentence of the abstract in [2] states quite plainly that “Deeper networks are more difficult to train” and this difficulty comes from “The deeper network has higher training error, and thus test error.” as stated in Figure 1 of the paper. This is by definition making a reference to underfitting although not explicitly stated. This definition of untrainable is supported by [3] which refers to the same underfitting in ResNet as “untrainable” in the introduction on Page 2. We explore residual connections in this paper with our experiment with a Deep CNN guided by a ResNet.
> > > >
> > > > The reviewer seems to have a different understanding of untrainable: a network is untrainable when it cannot fit to the training set at all. We designed an experiment for this. Suppose we take our target network and initialize all weights to 0. This is a great example of a network that cannot learn how to fit the training set. What if we use guidance in this setting? Even with CKA, this would still have activations that are set to 0 so CKA would have little effect. To resolve this, we add an arbitrary small noise perturbation to the activations on the first training step (1e-4) before continuing training or computing CKA.
> > > > We apply the zero-initialization with noise (Bad Init) to an RNN trained for copy-paste and Deep FCN trained for ImageNet as in the paper. Due to time constraints, we only use untrained guide networks to illustrate our point.
> > > >
> > > > |                | Baseline | Guide Accuracy |
> > > > |----------------|----------|----------------|
> > > > | RNN + Bad Init | 11.12    | 17.19          |
> > > > | RNN (Paper)    | 14.35    | 42.56          |
> > > >
> > > > |                     | Baseline Accuracy | Guide Accuracy |
> > > > |---------------------|-------------------|----------------|
> > > > | Deep FCN + Bad Init | 0.95              | 4.59           |
> > > > | Deep FCN (Paper)    | 1.65              | 13.10          |
> > > >
> > > > In this case, we still find improvements with guidance, although we do not recover the improvements reported in the paper. Injecting noise in the first training step is not effective to improve fitting when weights are initialized to 0 without guidance. We would like to point out that in this setting, we find that networks saturate immediately. We cannot share learning curves but the Deep FCN does not overfit but instead, immediately saturates. This shows that our initialization leads to a network that cannot improve its fit on the training set. In general, we hope this shows that our method is not misapplied. Representational alignment recovered a network in a very bad
> > > >
> > > > Would the reviewer’s concerns be alleviated if we did not use the term untrainable?
> > > >
> > > >
> > > > > Q2: "Inductive bias" from [5], thanks for pointing out this study, but it happens to prove my point in the weakness section. In the abstract of [5], "a randomly-initialized neural network can be used as a handcrafted prior",
> > > >
> > > > According to our understanding, the definition of handcrafted in this paper is different from handcrafted in the Gabor/Gaussian filter sense. The “Deep Image Prior” [4] paper defines a restoration objective in Equation 1 where every initial weight is randomly initialized and no weights are ever trained/pretrained. Therefore, handcrafted prior in this setting refers to designer‑chosen architecture. This is different from a Gabor-filter which uses fixed-scales and orientations. Gabor filters are a prior that is man‑made in the sense that a human explicitly specifies the architectural blueprint, even though the individual weights are not fixed ahead of time. [4] also presents a connection between the untrained network and the total variation (TV) norm, which is also handcrafted because a human chooses the function form of the TV-norm. It is important to point out that the untrained architecture and the TV-norm are not hand-crafted at the weight-level like a Gabor filter and their effects are not fixed across images.

---

> > > > > ### Author Response · Authors · 2025-08-09
> > > > >
> > > > > > Q2: My personal understanding on the model inductive bias, is the LEARNED features or patterns from a SPECIFICALLY DESIGNED network (in [5], they were tring to explore the LEARNED bias from a GENERATOR). Otherwise, if the main contribution in this study attempts to show an initialized network can also deliver inductive bias, I cannot say this is completely wrong, but this is quivalent to combining hand-crafted features with network together for a specific task.
> > > > >
> > > > > We would like to point out that our response to another reviewer shows that our results are not driven by a specific “handcrafted” architecture like using ResNet-18. As shown with Reviewer CY31, we can run guidance with many “handcrafted” CNNs and achieve similar results. Therefore, we are not specifically using a set of weights as with a Gabor filter. We believe this distinction is critical to the novelty of our work. In general, our work is not relying on a “handcrafted” guide network at all. We are choosing guide network architectures that empirically and canonically work for the given task.
> > > > >
> > > > > At a higher level, we would like to mention that handcrafting architecture is very methodologically different. The goal of our work is to specifically understand what should go into handcrafting an architecture. For example, including residual connections in comparison to using a standard Deep CNN. We know that [1] shows underfitting in Deep CNNs. Residual connections are a handcrafted design choice. Can we show that a Deep CNN without residual connections can get competitive performance to a network with residual connections? Can this happen from the architecture or does this need trained weights? We hope this shows why we believe the untrained guide setting is exciting -- it illuminates what architectural choices may actually matter and can be transferred from one network to another using representational alignment.
> > > > >
> > > > > We would like to emphasize our takeaways: We have shown that architectures that the community does not use due to optimization problems can have performance that is either significantly above chance or even competitive with architectures of today. This happens via representational alignment with a network that has a better architectural prior (handcrafted as the reviewer says but not in the same sense in our view) or better learned information. We believe this has significant potential to expand the architectures that people use in the community today as well as inform how we can handcraft architectures today. To us, guidance is a tool and tools are useful for enabling more discoveries. We believe these discoveries will allow the community to more easily pursue questions that the difficulties of training might have prevented in the past, especially with regards to understanding the relationships between handcrafted architectures.
> > > > >
> > > > > > Q5: "First, the focus of our paper is not only on overfitting." Sorry that I might mis-understood the focus, because the paper mainly associates "trainability" to "overfitting"(though this is questionable as discussed above)
> > > > >
> > > > > Thank you. We will aim to reduce the potentially strong association in our paper between trainability and overfitting.
> > > > >
> > > > > > Q6: I admit this would be a very interesting point to me, (very) loosely speaking, this is equivalent to learning from a less informative sub-space (model) to obtain something richer, but currently the draft didnot convincingly show this.
> > > > >
> > > > > We are glad that the reviewer found the results with smaller guide networks exciting. We find these results exciting as well. We believe this supports guidance. We refer to our argument on capacity for why we think this is intuitive as mentioned before. The target network does not have the capacity to learn useful information from a larger guide network like knowledge and structural priors. In the case of the RNN, more parameters are needed to learn the information in a transformer. We will dedicate more space in our paper to explaining this.
> > > > >
> > > > > Once again, we feel this falls under the scope of our paper very well. We can discover relationships between architectures that could not be explored before due to the difficulties of training i.e. how many parameters should an RNN have to better match a transformer.

---

> > > > > > ### Author Response · Authors · 2025-08-09
> > > > > >
> > > > > > >  Q7: Then this paper is the first to utlize the external hand-crafted features for training networks for down-stream tasks? I believe there was a time that this was a hot topic to explore. (just randomly searched one for reference[Local Learning with Deep and Handcrafted Features for Facial Expression Recognition])
> > > > > >
> > > > > > Thanks for the reference. We think the most important aspect of our paper is not just the using a “handcrafted” architecture, but the cross-architecture setting to get an architecture that is difficult to optimize to work much better.
> > > > > >
> > > > > > Once again, we reiterate that the connection to handcrafted features in our paper is at the level of architectural design. We will add these to related work as part of a discussion more broadly on how architectures can encode hand-crafted inductive biases and features. We believe the hand-crafted features referenced are broadly interesting but not the goal of our work, which aims to show how cross-architecture alignment can recover performance in a particular architecture that is difficult and our method can be used to explore what features can be transferred from one architecture to another.
> > > > > >
> > > > > >
> > > > > > [1] Wang and Fu. “Trainability Preserved Neural Pruning.” ICLR 2023.
> > > > > > [2] He et. al. “Deep Residual Learning for Image Recognition”, CVPR 2016.
> > > > > > [3] Peer et al. “Improving the Trainability of Deep Neural Networks through Layerwise Batch-Entropy Regularization”. TMLR 2022.
> > > > > > [4] Ulyanov et. al. “Deep Image Prior”. CVPR 2018.

---

### Official Review · Reviewer_iSbz · 2025-06-27

**Clarity:** 3
**Significance:** 3
**Originality:** 3
**Rating:** 5
**Confidence:** 3

**Summary:**

The paper proposes a new approach to boost the performance of neural networks with "wrong" inductive biases. The authors show that training a neural network (target network) while forcing its internal layers to behave (in terms of CKA alignment) like the layers of another neural network (the guide) with "proper" inductive biases. They show this brings about improvements even when the guide network is untrained.

**Questions:**

- Do initialization experiments give the same results as the distillation experiments (i.e., prior transfer term for first 300 step vs when the prior transfer term prolongs throughout entire training)? Is it true across language modeling too?

- What if you take the covariance matrices of the guide network's states rather than a more "complicated" loss term? Does it imply a better init for RNN?

**Ethical Concerns:**

["NO or VERY MINOR ethics concerns only"]

**Final Justification:**

This work helps to understand, from a clean point of view, novel aspects of teacher-student distillation. Especially, the role of just the network's bias on network distillation, at some points, just the prior gives superior results over trained models. This is exciting as a tool as well, as it allows to teach a model a bias unnatural to it. The authors have also discussed in the rebuttal that they were able to disprove the fact that initialization alone is enough, which is again a further contribution.

I like this paper for both its potential in applications as well as its novel approach to first-principles analysis of teacher-student dynamics.
My main concern is their presentation, conflating their contribution and distillation, and showing it from a slightly less principled angle than I'd wish them to.
A second concern is that there might be other explanations for the phenomenon, which is a main concern with any paper with such bold claims, but they have done more than enough for a single paper to avoid this criticism. It remains to be explored in future work.

**Limitations:**

yes

**Paper Formatting Concerns:**

- Line 176, what goes on there?

**Quality:**

3

**Strengths And Weaknesses:**

Strengths:
- The work is very creative! It shows that inductive biases can be copied to networks from the weights only, bypassing the rigidities of the architecture.
- It has a wide variety of experiments with a decent experimental design, including both image and language modeling. Showing good results across all of them.
- This opens the door for exciting approaches to initialization, potentially allowing the transfer of inductive tendencies of one hard-to-train architecture to a fast-to-train architecture.
- Remarkably, the approach does not require putting manual work into designing such initializations.


Weaknesses (minor):
- While, as I said, the approach is very novel as far as I can tell, the authors do not make the distinction between THEIR contribution (the transfer of inductive prior only) and ESTABLISHED approaches (teacher-student) and consider them together. I would make the distinction clearer. Having said that, it is mostly a presentation matter, as it is clear that the method works in the important case (i.e., when the guide is untrained)

- Intiuition: I would have liked it if the authors could explain the mechanisms as far as they can (beyond the abstract interpretation of transfer of prior). For example, I think it is worthwhile to mention in passing that in the case of FCN -> ResNet, the inductive bias is probably a simple one (probably making the initialization matrix sampled from around the identity matrix). Other things I am curious about:
* Intuition about how CKA can encode complicated information such as the inductive bias of a network (eg, how would the CKA of an untrained residual network differ from not untrained FCN, is it a matter of covariance matrices?)


- Finally, I think that the initialization interpretation is somewhat more satisfying and less unsettling than that that requires training with the inductive bias throughout the entire training process.

---

> ### Author Rebuttal · Authors · 2025-07-30
>
> Thank you for your thoughtful review. We are glad that the reviewer found the paper creative, appreciated the wide variety of experiments, and was excited by the initialization experiments. We hope to address concerns here:
>
> > W1: the authors do not make the distinction between THEIR contribution (the transfer of inductive prior only) and ESTABLISHED approaches (teacher-student) and consider them together
>
> Thanks. We will make this clearer in the paper. We agree that methodologically, guidance makes a contribution with transferring the inductive prior only. However, we also want to point out that we can use guidance to differentiate between transferring knowledge and transferring a prior. All experiments with an untrained guide transfer an architectural prior such as translational equivariance or receptive fields in a convolutional layer. When we transfer knowledge, we use a trained guide similar to teacher/student settings. Guidance can identify when the architectural prior is useful for improving task performance, a finding that has useful scientific implications. Knowledge distillation cannot do this -- when we use an untrained network as a teacher, we find performance decreases, even in the initialization case (see below). This is intuitive; we are matching outputs of a randomly initialized network on noise. But, beyond the simple methodological difference, this is the scientific setting that guidance can distinguish. We believe such distinctions can help us understand relations between architectures i.e. which networks can take advantage of a structural prior to improve task performance rather than requiring the same structural prior along with knowledge.
>
> > W2: if the authors could explain the mechanisms as far as they can (beyond the abstract interpretation of transfer of prior). For example, I think it is worthwhile to mention in passing that in the case of FCN -> ResNet, the inductive bias is probably a simple one (probably making the initialization matrix sampled from around the identity matrix).
>
> Thank you. Yes, we agree that we would like to go deeper into understanding guidance. The extra CKA term reshapes the optimization landscape by pulling the FCN’s activations toward the manifold realized by a frozen ResNet. Because CKA depends only on gram matrices, the FCN does not need to learn literal identity weight matrices; instead, it must recover the ResNet’s locality and weight-sharing statistics.
>
> There are many additional results that support this possibility. Appendix I shows that, after guidance, the FCN’s layerwise ID curve is similar to the ResNet’s. High ID delays collapse that normally indicates memorization, acting as an implicit regularizer. This is also supported by initialization experiments. Furthermore, Appendix K demonstrates that guiding early ResNet layers delivers the largest performance boost. Those layers encode exactly the low-level locality and translation equivariance biases that an FCN lacks. This is supported by the error consistency metrics. Overall, this connects with the ID findings: the convolutional geometry injected by those layers maintains high ID and eases generalization.
>
> The ResNet’s skip connection does cause its first few activations to resemble the input, so matching them encourages near-identity activations in the FCN. But our intuition is that the performance jump comes from the richer structure supplied by convolution and weight sharing, not from setting FCN weight matrices to the identity. Guidance works because it steers the target network into a representational regime that combines statistically useful activations (high ID, initialization) and functional geometric priors (locality, equivariance).
>
> > W3: Intuition about how CKA can encode complicated information such as the inductive bias of a network (eg, how would the CKA of an untrained residual network differ from not untrained FCN, is it a matter of covariance matrices?)
>
> This is a great question. CKA is a measure that depends on second-order statistics, specifically pairwise sample distance matrices i.e. the gram matrices formed by taking a distance between every pair of samples with shape [num_samples x num_samples]. Architectural choices imprint distinct features on those statistics. For example, consider local receptive fields in a convolutional layer. Units that cover neighbouring pixels receive correlated input, and this is reflected in our activations. Such correlations are reflected in our distance matrices, and these can be transferred to distance matrices associated with FCN layers that lack local correlations. Similarly, weight sharing, where the same kernel is applied at every spatial location, will also be reflected in a distance matrix, even if weights are random.
>
> > Q1: Do initialization experiments give the same results as the distillation experiments (i.e., prior transfer term for first 300 step vs when the prior transfer term prolongs throughout entire training)? Is it true across language modeling too?
>
> While we cannot share external links to images of plots, we have run an experiment where we recreate our initialization with a distillation loss function rather than guidance. Our finding is that distillation does not recreate the finding with initialization. This is expected: we are matching outputs of a randomly initialized network on noise. With distillation, our initialization experiment gets an accuracy of 1.75%, which is close to running the Deep FCN with standard training.
>
> We find that initializing an RNN with guidance leads to little improvement in language modeling or copy-paste. This is useful for confirming the specific failures of the RNN: initialization does not solve problems with repeated matrix multiplications that must learn to remember, mix, or forget inputs from prior time steps. We believe this is unlikely to have been solved using better initialization since layer errors compound in a deep RNN. We believe the initialization experiments are better reserved for the FCN and are excited to try extending them to transformers as well.
>
> > Q2: What if you take the covariance matrices of the guide network's states rather than a more "complicated" loss term? Does it imply a better init for RNN?
>
> To ensure our own understanding, it seems the reviewer is asking whether we could replace CKA with a loss function that computes the L1 or L2 distance between the guide network and target network covariance matrices for a layer pair. It’s likely that this could transfer some architectural bias. But, our intuition is that it would be harder to optimize as it gives up certain properties of CKA such as scale- and rotation-invariance. Scale invariance is important because if one layer’s variance spikes, this would dominate the basic covariance loss. Similarly, matching covariance matrices would force an exact orientation match due to a lack of rotation invariance. And, CKA implements an implicit layer normalization while covariance matching does not, meaning that deeper layers with a tiny variance would contribute little to the covariance matching loss. However, we hope to investigate many possible alignment functions, from ones with more degrees of freedom to ones with less degrees of freedom to better understand how these properties influence how information is shared across networks. We also note that the memory usage from the covariance matrices (num_features x num_features) would be significant.
>
> As discussed above, we don’t believe that guidance would be a very good initialization for the vanilla RNN due to the specific failures of RNNs themselves.
>
> > Q3: Line 176, what goes on there?
>
> Thank you for catching this. Apologies, this was typo that was probably from shortening the paper. The sentence should say "To systematically evaluate our approach, we divide our experiments into settings with untrainable architectures and untrainable tasks."

---

> ### Comment · Reviewer_iSbz · 2025-08-05
>
> Thanks for addressing my questions and concerns so thoroughly (with somewhat too heavy technicality for my expertise in the field). Many interesting and thought-out directions.
> I will address the points mentioned and summarize what I hope to see in the final version.
>
> About W1, this is very important. Though it is only a presentational problem, it is highly important for organization. I feel that the response espoused my claims. IIUC, the authors say that both the trained variant and untrained variant are relevant in certain cases, which I agree. Presentationally, you still need separation. Admission of this being a form of Teacher-Student distillation is super important to have. You should focus on cases where prior-only distillation is beneficial and even superior. You can show the other cases, but I won't focus on them.
>
> About Q1, super interesting, not trivial and kind of contradictory to my intuition! Though I get the reasoning but not at all my guess at first. Should probably be in the paper!
>
> About Q2, from my experience, many past works that have very complicated losses boil down to at best matching second-order statistics. I would look have looked into it some more (not necessarily here) to get a cleaner result.
> Maybe you're right that the invariances you discuss cannot be (even approximately?) encoded in gram matrices, but are we sure they are transfered here?
>
> There is still much intuition missing as to what (really) happens beyond the scenes and of course it might be underwhelming in the end. It is this reason that I hope to see a (slightly underwhelming) explanation in the form of matching second order statistics, in fear of some unknown, very underwhelming explanation.
> Because you might agree with me, that it is still mechanistically not understandable how matching the output vector can encode the more complicated priors (I can convince myself for rotation invariance, but it's also a less interesting case anyways).
>
> Anyways, hope any of this helps.

---

> ### Author Response · Authors · 2025-08-08
>
> We thank the reviewer for their suggestions.We will certainly be sure to frame the paper properly and be sure to include newer results that were requested here.
>
> Regarding presentation, we believe the contribution of our paper goes beyond the methodology and this is precisely why we argue that the cases where trained guide networks work are interesting. We aim to cast guidance as a conceptual lens. When we use an untrained guide network, we reveal what an architecture by itself brings to the table. Guidance with a trained guide reveals how much learned representations can change performance. If we were to remove the second aspect from the paper, we would reduce guidance to an “odd regularizer.” If we remove the first, we fall back into teacher-student comparisons. The novelty lies precisely in showing that the same loss function augmentation can shift between these two regimes, giving researchers a dial for disentangling structure from learning. In that sense, the distinction resides inside guidance. Separating the cases would obscure that dial.
>
> We wanted to follow up on the discussion of using second order statistics (covariance matrices of shape [num_features x num_features]) as our measure of representational alignment rather than CKA. We first run a comparison between matching second-order statistics rather than using CKA. We paste results below for our Deep FCN setting and RNN with language-modeling setting.
>
> | Experiment             | Untrained Guide ImageNet Accuracy | Trained Guide ImageNet Accuracy |
> |------------------------|-----------------------------------|---------------------------------|
> | DeepFCN + Second Order | 5.30                              | 5.11                            |
> | DeepFCN + CKA          | 7.50                              | 13.10                           |
>
> | Experiment         | Untrained Guide Test Perplexity | Trained Guide Test Perplexity |
> |--------------------|---------------------------------|-------------------------------|
> | RNN + Second Order | 68.16                           | 55.66                         |
> | RNN + CKA          | 59.61                           | 40.01                         |
>
> In general, we find that matching second-order covariance matrices does not provide improvements as substantial as those with CKA. This partially supports the possibility that the invariances in CKA are useful and necessary to obtain the performance improvement seen in guidance.
>
> We want to emphasize that we believe that, like the proposed loss from the reviewer, using CKA leads to a simple loss as well. CKA matches second-order statistics using distances across samples. The main goal of CKA is to measure whether the distance patterns between points in two sets of representations are similar. As a metric for alignment and similarity, this is both weak and simple. We don’t believe that there is a methodologically underwhelming story that will arise due to using a very complex approach for our alignment. Of course, CKA has certain, simple invariances, as discussed before. Our future scope can focus on which invariances in CKA are useful for letting information travel across two architectures. We also hope to go deeper into architectural representations when training with guidance versus training without guidance such as studying the spectral distribution, sparsity, etc. We could plan to study the optimization landscape in more detail. There may exist more rich alignment functions that are more complicated that lead to even stronger results, performance-wise. The simplicity of our method allows us to consider these questions.
>
> These threads demonstrate the major possibility for future work in this space. Guidance is a tool, not a finished and well-understood theory or doctrine. Tools are useful for enabling more discoveries. We believe these discoveries will allow the community to more easily pursue questions that the difficulties of training might have prevented in the past, especially with regards to understanding the relationships between architecture.

---

### Official Review · Reviewer_CY31 · 2025-07-02

**Clarity:** 2
**Significance:** 2
**Originality:** 3
**Rating:** 4
**Confidence:** 4

**Summary:**

This paper introduces guidance, a novel method to train architectures typically considered untrainable for certain tasks by transferring inductive biases from a guide network. The key idea is to steer a target network using representational alignment (e.g., CKA) with the guide. The target network is trained to minimize both the task loss and a representational similarity loss relative to the guide. The authors demonstrate improvements in training performance across architectures and tasks, for example, improving RNNs using Transformers, and mitigating overfitting in FCNs using ResNets. The paper includes empirical analysis to differentiate architectural and learned inductive biases, highlights cases where guidance helps (or doesn’t), and proposes practical applications such as new initialization strategies and architecture search.

**Questions:**

- Can the authors provide theoretical evidence on why representational alignment to a guide helps avoid overfitting/underfitting, beyond anecdotal examples?

- What are the memory and runtime costs of this method compared to training a standard network or distillation-based approaches?

- Why not use the guide network directly for inference, especially if it already performs well on the task?

- How sensitive is the method to the choice of guide network? Is there a risk that a poor guide could hurt performance?

**Ethical Concerns:**

["NO or VERY MINOR ethics concerns only"]

**Final Justification:**

My concerns are almost resolved, except a rigorous theoretical justification for the proposed method.

**Limitations:**

Yes!

**Paper Formatting Concerns:**

The format of the paper is fine.

**Quality:**

2

**Strengths And Weaknesses:**

*Strengths*

- The guidance mechanism presents a novel and creative way to inject inductive bias into architectures that struggle to train successfully on specific tasks.

- The authors explore a range of architectures (RNNs, Transformers, FCNs, CNNs), tasks (e.g., copy-paste, image classification), and guide/target combinations to support their claims.

- The method could be influential in improving training robustness and re-enabling the use of previously discarded architectures.

*Weaknesses*

- My main concern of this paper is the lack of theoretical justification of the method. The paper does not provide a clear theoretical grounding for why or how guidance helps prevent overfitting or underfitting in the target network. The mechanism by which representational similarity to guide network results in better generalization is not well understood or analyzed.

- Another concern, that may prevent the method to be useful in practice, is the high memory and computation cost of the method. The method involves keeping the guide network (often a large model) in memory during training, and performing additional gradient computations for the representational loss. This raises concerns about scalability, especially for large guide networks.

- The paper does not justify why the target network is trained to mimic the guide rather than simply using the guide network itself for the task. In some cases, using the guide directly may be simpler and more efficient, especially if the guide is already trained.

---

> ### Author Rebuttal · Authors · 2025-07-30
>
> We thank the reviewer for their thoughtful review. We are glad the reviewer found the paper novel and interesting, appreciated the wide variety of settings, and considered guidance to be potentially influential. We hope to address some concerns here.
>
> > W1: theoretical justification of the method. The paper does not provide a clear theoretical grounding for why or how guidance helps prevent overfitting or underfitting in the target network
>
> Thank you for this question. While we don’t have theoretical justification in the form of a proof, we do have potential explanations that reference prior work. We will update the draft with this discussion and will add these references as related work.
>
> First, [1] is a new paper that considers task-aware representational alignment. Their theory provides a generalization bound via kernel alignment. They show that when a “stitcher” maps representations or a source network to a target output, the excess risk of the stitched model is upper-bounded by the CKA alignment between them. This provides a learning-theoretic guarantee that the CKA term in guidance reduces the hypothesis class possibilities seen by an optimizer. Overfitting or underfitting becomes harder.
>
> [2] investigates how a network’s neural tangent kernel (NTK) aligns with a target output during training. The paper shows that NTK alignment accelerates convergence and lowers generalization error in deep linear networks. This aligned kernel condition is inserted by hand in guidance. Similarly, [3] uses Rademacher complexity tools to show that alignment of tangent-kernel features onto a small set of task-relevant directions compresses the effective model class. This formalizes the notion of guidance as an automatic regularizer, where task directions are replaced by the guide network settings.
>
> Finally, [4] demonstrates that after training, the top singular vectors of a network’s hidden activations align with the task target vectors. This empirically supports the layerwise CKA choice in guidance.
>
> A full PAC-style proof specialized to guidance has not been shown in our paper. But we hope that the above covers a theoretical gap. We believe that CKA bounds the risk or complexity in terms of kernel alignment. The NTK and Rademacher analyses show that alignment shrinks the effective hypothesis space and improves conditioning. This aligns with findings based on singular vectors. We could sharpen this theory by changing the alignment used in guidance e.g. moving from aligning on kernels to aligning on singular vectors or eigenvectors instead.
>
> > W2: the high memory and computation cost of the method
>
> This is a fair point. However, one of the exciting findings in our paper is that guide networks need not be too large. In fact, we find that guide networks can be an order of magnitude smaller in terms of the number of parameters and still achieve reasonable results. For example, we tried further experiments where we designed an RNN with 345M parameters (equivalent to GPT-2 Medium) and demonstrated that competitive scaling using an untrained transformer guide that had 50M parameters. We are experimenting with even smaller transformer guides and expect to see similar findings. The smaller the guide, the less memory the model and activations take.
>
> We would also like to point out that in the current compute paradigms e.g. test-time-compute, inference costs are much more expensive than training costs. Having networks with better inference time efficiency is important. Guidance may come with a higher training cost, but the result is a network that has much lower inference cost. This is important given that the current finding with test-time-compute is that generating more tokens leads to better performance.
>
> Guidance can also work naturally with data parallelism, tensor parallelism, or model parallelism setups in most deep learning libraries. We have used some of these techniques for scaling up RNNs. The main footprint memory-wise is storing activations, but there are many techniques one could use to avoid taking GPU memory, such as storing activations to disk or using half-floats instead of full floats. The main speed footprint comes from CKA calculations, but there are parallelism optimizations that can be applied.
>
> > W3: The paper does not justify why the target network is trained to mimic the guide rather than simply using the guide network itself for the task
>
> We apologize, we should have made this clearer. As stated above, one main setting our paper focuses on is when inference costs are much larger than training costs e.g. with test-time-compute. Models like transformers, are slow at inference time, with quadratic time attention. This severely impedes the deployment efficiency of current sets of models. We expect the trend of inference costs dominating training costs to continue. This motivates the need for novel, efficient architectures with more reasonable inference costs. Our paper aims to expand this potential space of architectural selection choices. There are many efforts that explore alternative approaches, such as SSMs/Mamba, and we believe guidance can aid this effort to create more efficient architectures. With cases like RNNs or FCNs, we can’t guarantee that guidance will find a target network architecture with better efficiency; efficiency is not always about making FLOPs parallelizable as done in training but ensuring that more FLOPs can be used more quickly, as seen in test-time-compute.
>
> Furthermore, the findings in our paper could be very beneficial for expanding possible network selection for practitioners when searching for network architectures as an alternative to neural architecture search. One can now design architectures with better guarantees on training and can compare architectures more fairly without the need for evaluating conditions on trainability, as done in NAS. In general, a larger number of potential architectures leads to more settings where networks can be applied.
>
> > Q1: What are the memory and runtime costs of this method compared to training a standard network or distillation-based approaches?
>
> We present our runtime and memory costs for two settings. For the Deep FCN setting, our base memory usage is 8GB (batch size of 256) during training and our wallclock runtime is about 17 minutes per epoch on an H100 with our current implementation. With an untrained ResNet-18 guide, our memory usage is 10GB of GPU memory and our runtime is about 28 minutes per epoch on an H100 with our current implementation. Similarly, for our RNN with copy-paste setting, our base memory usage is 2GB of memory during training and our wallclock time is about 3 minutes an epoch on an H100 with our current implementation. With an untrained transformer guide, our memory usage is 3GB an epoch and our wall clock time is also about 3 minutes an epoch. We believe this shows that the memory and runtime cost of guidance is not much more expensive than training a standard network. As stated previously, this training cost can come with benefits for the larger inference cost.
>
> > Q2: How sensitive is the method to the choice of guide network? Is there a risk that a poor guide could hurt performance?
>
> The method is sensitive to the choice of guide network from running a few sanity checks like guiding a Deep FCN with another Deep FCN. In this case, we find that the Deep FCN overfits and does not improve performance. We also find that using larger guide networks may not necessarily improve the performance of the target network as much due to capacity issues in the target network. The target network may not be expressive enough to learn the representation in the guide network. We will add these discussions to the paper.
>
> [1] Insulla et. al. “Towards a Learning Theory of Representation Alignment”. Preprint, 2025.
>
> [2] Shan and Bordelon. “A Theory of Neural Tangent Kernel Alignment and Its Influence on Training”, Preprint. 2022.
>
> [3] Baratin, George, et. al. “Implicit Regularization via Neural Feature Alignment”. AISTATS, 2021.
>
> [4] Imani, et. al. “Representation Alignment in Neural Networks”, TMLR 2023.

---

### Note · Authors · 2025-08-14

Dear all,

We thank the reviewers for their service in reading our paper and considering our responses. We are glad we could strengthen the confidence of reviewers who were initially positive and resolve concerns from reviewers who were initially negative. Our paper recovers performance of networks that have difficult optimization failures using representational alignment with another architecture. We want to highlight some of our results, such as findings with vanilla RNNs, which improve when guided by untrained transformers even in language modeling. We hope this shows the possibility to train on many types of network architectures that were previously considered difficult to optimize e.g. Sparse MoE. Our findings regarding initialization open new possibilities for promising algorithms compared to prior work that uses random distributions. We summarize our additions below:

* Underlying mechanisms for guidance [CY31, xZWb, iSbz]: We thoroughly considered and addressed many concerns related to theoretical motivations for guidance. We referenced many papers supporting our findings and ran related experiments to investigate our loss function with CKA in comparison with other ways of doing second-order statistics. We clarified the simplicity of our method.
* Trainability and novelty [Tqwd]: We clarified our setting and use of the word “untrainable” in the paper. This paper focuses on an applied setting which considers networks which overfit, underfit, or don’t incorporate memory.  We expanded guidance’s applicability by considering a network that was even more difficult to recover performance under. We showed guidance could recover performance in this case.
* Baselines [xZWb, Tqwd]: We demonstrated that guidance performs better than other regularization methods like L2 or better optimizers like AdamW. We also demonstrated that guidance performs better than distillation.

We want to emphasize that our paper is an exciting start to a long line of work that could recover many architectures that are useful for the new paradigm of increased inference time compute. We also see many possibilities for understanding deep connections between architectures, something that is still unknown. We plan to dedicate significant future work to understanding guidance, taking it from a tool as used here, to a truly understandable mechanism. We are excited about extending our line of work, and we believe the community will be as well.

We thank you all for your time,

Authors

---

### Decision · Program_Chairs · 2025-09-17

**Decision:**

Accept (poster)

**Comment:**

This paper introduces guidance, a method for training networks that are not considered suitable for certain tasks by utilizing a guide network. The method introduces a novel way to incorporate inductive bias into the training process. There were concerns with the memory and computational costs of the method as well as the lack of theory behind some of the claims. Overall, the reviews were positive and I believe the paper should be accepted. The authors should consider removing theoretical claims as the paper is mostly empirical.